# Developmental atlas of phase-amplitude coupling between physiologic high-frequency oscillations and slow waves

Kazuki Sakakura [1,2,3], Naoto Kuroda[1,4], Masaki Sonoda[1,5], Takumi Mitsuhashi [1,6], Ethan Firestone [1,7], Aimee F. Luat[1,8,9], Neena I. Marupudi[10], Sandeep Sood[10] & Eishi Asano [1,8] ✉

We investigated the developmental changes in high-frequency oscillation (HFO) and Modulation Index (MI) – the coupling measure between HFO and slow-wave phase. We generated normative brain atlases, using subdural EEG signals from 8251 nonepileptic electrode sites in 114 patients (ages 1.0–41.5 years) who achieved seizure control following resective epilepsy surgery. We observed a higher MI in the occipital lobe across all ages, and occipital MI increased notably during early childhood. The cortical areas exhibiting MI co-growth were connected via the vertical occipital fasciculi and posterior callosal fibers. While occipital HFO rate showed no significant age-association, the temporal, frontal, and parietal lobes exhibited an age-inversed HFO rate. Assessment of 1006 seizure onset sites revealed that z-score normalized MI and HFO rate were higher at seizure onset versus nonepileptic electrode sites. We have publicly shared our intracranial EEG data to enable investigators to validate MI and HFO-centric presurgical evaluations to identify the epileptogenic zone.

In patients with drug-resistant focal epilepsy, surgical resection of the epileptogenic zone results in long-term seizure control, and clinicians employ intracranial EEG (iEEG) recording via subdural or depth electrodes to localize such areas. The occurrence rate of spontaneous interictal high-frequency oscillation (HFO) - defined as transient bursts of ≥ 80 Hz activity[1] - has garnered attention as an iEEG-based epilepsy biomarker, since it is generally increased in the seizure onset zone (SOZ)[2]. A randomized clinical trial reported that the efficacy of intraoperatively measured HFO-guided resection in controlling seizures was non-inferior to that of conventional resection guided by interictal spike discharges in patients with extra-temporal lobe epilepsy but not in those with temporal lobe epilepsy[1].

HFO are generally nested within slow waves during sleep, and modulation index (MI) is a quantitative measure of the coupling strength between HFO amplitude and slow-wave phase[3,4]. MI represents a potentially valuable iEEG-based epilepsy biomarker; in particular, it is an effective proxy measure for spike-and-wave discharges, since each spike component is associated with a transient increase in HFO that is stereotypically coupled with a delta wave$_{3-4Hz}$[5,6]. Retrospective analysis of iEEG recordings from epilepsy patients demonstrated that MI was elevated in the SOZ, and complete resection of

[1]Department of Pediatrics, Children's Hospital of Michigan, Detroit Medical Center, Wayne State University, Detroit, MI 48201, USA. [2]Department of Neurosurgery, Rush University Medical Center, Chicago, IL 60612, USA. [3]Department of Neurosurgery, University of Tsukuba, Tsukuba 3058575, Japan. [4]Department of Epileptology, Tohoku University Graduate School of Medicine, Sendai 9808575, Japan. [5]Department of Neurosurgery, Yokohama City University, Yokohama-shi 2360004, Japan. [6]Department of Neurosurgery, Juntendo University, Tokyo 1138421, Japan. [7]Department of Physiology, Wayne State University, Detroit, MI 48201, USA. [8]Department of Neurology, Children's Hospital of Michigan, Detroit Medical Center, Wayne State University, Detroit, MI 48201, USA. [9]Department of Pediatrics, Central Michigan University, Mount Pleasant, MI 48858, USA. [10]Department of Neurosurgery, Children's Hospital of Michigan, Detroit Medical Center, Wayne State University, Detroit, MI 48201, USA. ✉e-mail: easano@med.wayne.edu

high-value MI sites was associated with postoperative seizure control[5,7]. Advantages of MI over HFO rate as a biomarker include being continuous and requiring less computational time[7].

In clinical practice, caution is required when using MI and HFO rate as iEEG biomarkers because these signals are endogenously present and exhibit topographical variation across the cortex[8–10]. The occipital lobe is an area that naturally exhibits elevated MI and HFO rate, regardless of epileptogenicity[7,11–13]. However, the developmental implications of these physiologic distributions remain unknown, and gleaning this knowledge is expected to improve interpretation of the iEEG biomarkers' significance. In this study, we created normative atlases visualizing the developmental changes in MI and HFO rate across nonepileptic iEEG electrode sites. To optimize the generalizability of our observations, we employed four different detector toolboxes to calculate HFO rate[14–17].

Invasive studies of healthy cortices in animal models and nonepileptic cortices of patients with focal epilepsy have reported that spontaneous HFO are nested in delta waves and involve large-scale cortical networks during slow-wave sleep[11,18–22]. This nested HFO during slow-wave sleep is believed to replay sensory-related neural communications across two regions occurring during wakefulness and facilitate the consolidation of long-term memory related to given sensory representations. Some hypothesize that spontaneous $HFO_{\geq 80\ Hz}$ nested in delta $waves_{0.5-1Hz}$ in the primary visual cortex during slow-wave sleep may be involved in the consolidation of visual memory[11,23–26]. Given that preverbal infants are known to encode and retain visual information and this process likely involves delta-nested HFO during slow-wave sleep[27–29], we hypothesized that even in young children, the spatial profile of delta-nested HFO would already be developed. Specifically, we expected to see physiological enhancement of HFO-delta phase-amplitude coupling in the nonepileptic occipital lobe, as is typically observed in older children and adults. Since pattern recognition memory is known to improve most rapidly before six years of age[30], we hypothesized that the developmental slope of HFO-delta phase-amplitude coupling would be steepest during young childhood.

Previous studies in animal models and patients with focal epilepsy have demonstrated that neural circuits engaging in high-frequency cortical activity are susceptible to use-dependent modifications of synaptic transmission that enhance effective connectivity via monosynaptic white matter tracts in the developing brain[31–33]. White matter development, as rated by MRI tractography, is generally drastic during early childhood and modest during late adolescence[34,35]. We thus visualized white matter pathways supporting the development of neural communications via HFO nested in delta waves during slow-wave sleep. To this end, we incorporated MRI tractography and identified the white matter tracts directly connecting cortical regions with significant developmental co-growth of HFO-delta phase-amplitude coupling.

In this study, our normative atlases reveal a marked enhancement of MI in the nonepileptic occipital lobe compared to the other lobes across childhood (Supplementary Fig. 1). Occipital MI shows an age-related increase particularly during early childhood, with the vertical occipital fasciculus and posterior callosal fibers substantiating this developmental co-growth of MI across occipital cortices. In contrast, the HFO rate in the nonepileptic occipital area exhibits no significant correlation with age, whereas each of the other three brain lobes demonstrates an inverse relationship with age. Electrode sites within the seizure onset zone show significant, upward deviations in MI and HFO rate exceeding the normative values expected for given regions and ages. These findings suggest that our normative atlas may serve as a valuable reference in the context of epilepsy presurgical evaluations.

## Results
### Patients
We studied 114 patients (ages 1.0–41.5 years) who met the eligibility criteria (Table 1; Supplementary Table 1; Supplementary Fig. 2). A total

## Table 1 | Patient Profile

| | |
|---|---|
| Number of patients | 114 |
| Mean age in years (range) | 11.4 (1.0–41.5) |
| Proportion of female (%) | 47.4 |
| Sampled hemisphere (%) | |
| Left | 45.6 |
| Right | 43.0 |
| Both | 11.4 |
| Seizure onset zone (%) | |
| Frontal | 24.6 |
| Temporal | 47.4 |
| Parietal | 27.2 |
| Occipital | 19.3 |
| MRI-visible structural lesion (%) | 66.7 |
| Mean number of antiseizure medications (range) | 2.0 (1–5) |

Supplementary Table 1 presents the distribution of patient ages.

of 8251 artifact-free nonepileptic electrode sites (mean: 72.4 per patient; range: 12–121; Fig. 1) were available for generating normative atlases. In addition, 1045 electrode sites within the SOZ (mean: 8.8 per patient; frontal: 260; temporal: 418; parietal: 243; occipital: 85) were available to assess the utility of HFO and MI in distinguishing SOZ from nonepileptic electrode sites. Supplementary Fig. 3 explains how the 8251 artifact-free nonepileptic electrode sites satisfied the eligibility criteria.

### Slow-wave sleep
The Rayleigh's test indicated a significant deviation from a uniform distribution in the onset time of studied slow-wave sleep epochs (z-value: 83.5; $p < 0.001$), with the peak onset time observed at 1:00 am. We found no significant correlation between patient age and the onset time of the studied slow-wave sleep epoch (p-value: 0.40 using Spearman's rank test).

### Developmental atlas of cortical MI
The violin plots illustrate the developmental changes in $MI_{\geq 80\ Hz\ \&\ 0.5-1\ Hz}$ at given lobes (Fig. 2); thereby, $MI_{\geq 80\ Hz\ \&\ 0.5-1\ Hz}$ denotes the strength of coupling between the amplitude of $HFO_{\geq 80\ Hz}$ and the phase of slow-$wave_{0.5-1\ Hz}$. Supplementary Movie 1 provides a comprehensive summary of normative development of $MI_{\geq 80\ Hz\ \&\ 0.5-1\ Hz}$, estimated by univariate regression models with age, square root of age ($\sqrt{age}$), or log10 age as independent variables. Linear and nonlinear regression models consistently indicate significant developmental growth in $MI_{\geq 80\ Hz\ \&\ 0.5-1\ Hz}$ prominently in the occipital lobe during young childhood and beyond. The assessment of Akaike Information Criterion (AIC) suggests that the developmental growth in occipital $MI_{\geq 80\ Hz\ \&\ 0.5-1\ Hz}$ was drastic during young childhood and modest during older childhood. The mean regression slope was +0.0072/year (95%CI: +0.0050 to +0.0093; AIC: −368.0) in the linear regression model, +0.047 /$\sqrt{year}$ (95%CI: +0.034 to +0.060; AIC: −374.3) and +0.14/$\log_{10}$ year (95%CI: +0.10 to +0.18; AIC: −376.6) in the nonlinear models. The regression slopes of $MI_{\geq 80\ Hz\ \&\ 0.5-1\ Hz}$ in the parietal, temporal, and frontal lobes were much flatter, compared to the occipital lobe (Fig. 2). Table 2 provides the regression slope, uncorrected p-value, t-value, and degrees of freedom (DF) in a model incorporating $\sqrt{age}$ as an independent variable. The overall findings mentioned above remained qualitatively similar when excluding three patients over 21 years old from the analyses (Supplementary Table 2).

As detailed in Supplementary Fig. 4 and Supplementary Movie 2, the developmental changes of $MI_{\geq 80\ Hz\ \&\ 0.5-1\ Hz}$ and $MI_{\geq 80\ Hz\ \&\ 3-4\ Hz}$ were qualitatively similar. The developmental growth of $MI_{\geq 80\ Hz\ \&\ 3-4\ Hz}$ was

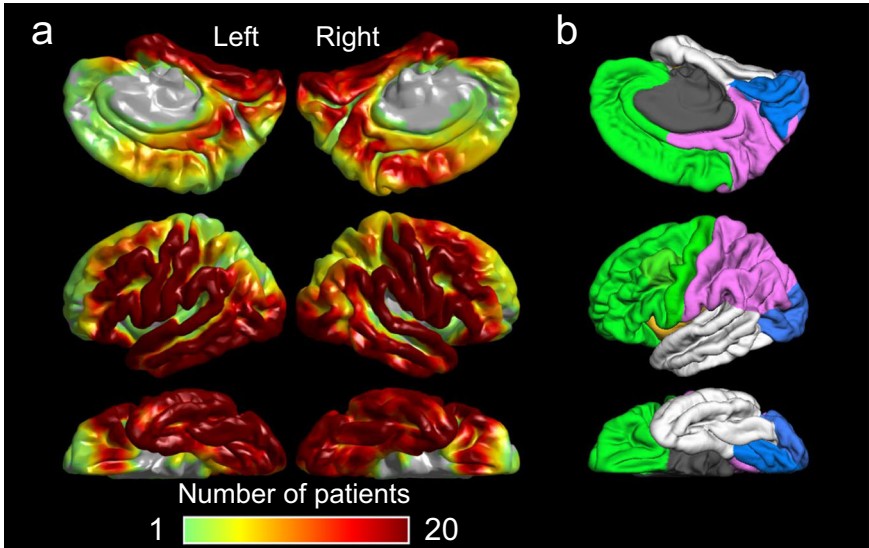

**Fig. 1 | Spatial distribution of intracranial electrode sampling. a** The figure shows the number of patients whose artifact-free, nonepileptic intracranial EEG data were available at each site. **b** The cerebral cortex was divided into four lobes using FreeSurfer (https://surfer.nmr.mgh.harvard.edu/fswiki/CorticalParcellation). Green represents the frontal lobe, white represents the temporal lobe, pink represents the parietal lobe, and blue represents the occipital lobe.

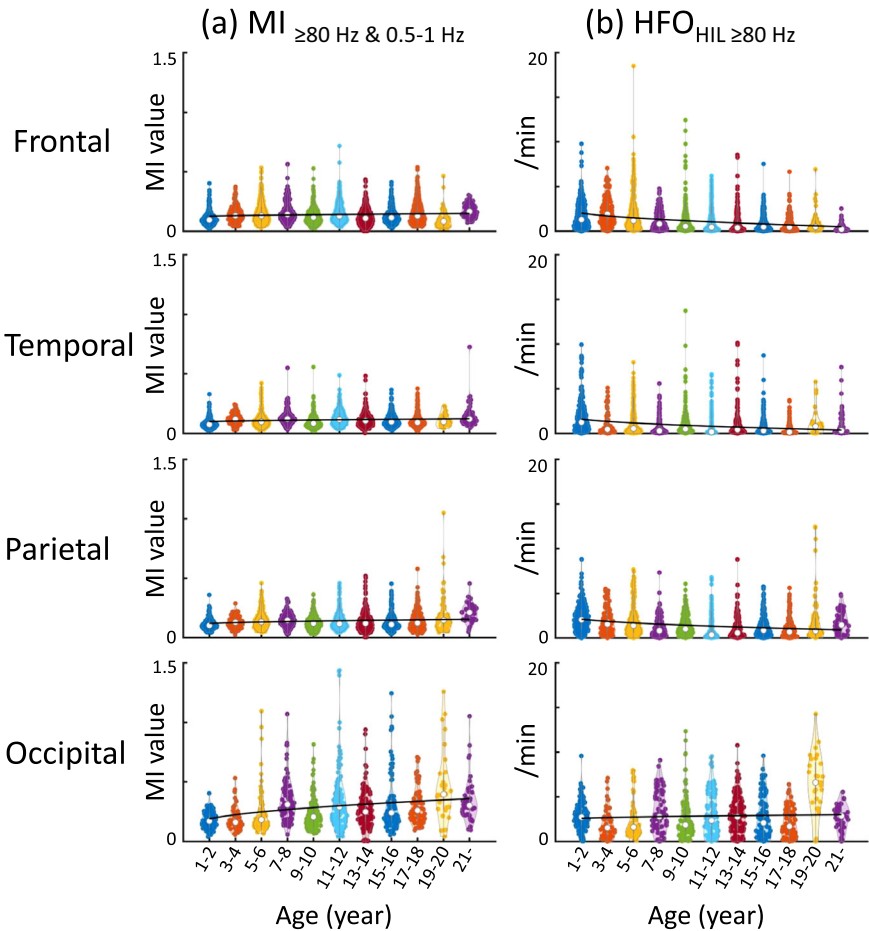

**Fig. 2 | The developmental changes of cortical MI and HFO at given lobes.** **a** $MI_{\geq 80 Hz \& 0.5\text{-}1 Hz}$ denotes the strength of coupling between the amplitude of $HFO_{\geq 80 Hz}$ and the phase of slow-wave$_{0.5\text{-}1 Hz}$, as rated by modulation index. **b** Occurrence rate (/min) of $HFO_{HIL \geq 80 Hz}$ as defined by the Hilbert method. In each violin plot, a regression line is provided based on a model incorporating the square root of age ($\sqrt{age}$) as an independent variable. The white circle within each violin plot represents the median. MI Modulation index. $MI_{\geq 80 Hz \& 0.5\text{-}1 Hz}$ denotes the strength of coupling between the amplitude of $HFO_{\geq 80 Hz}$ and the phase of slow-wave$_{0.5\text{-}1 Hz}$. $HFO_{HIL}$ High-frequency oscillation defined by the Hilbert method.

highest in the occipital lobe. The mean regression slope was +0.019/√year (95%CI: +0.014 to +0.023) in the occipital lobe but not significantly different from zero in the remaining lobes. Figure 3 displays snapshots of normative developmental atlases for MI$_{\geq 80\ Hz\ \&\ 0.5\text{-}1\ Hz}$, emphasizing the increased values in cortical regions near the calcarine sulcus.

## Developmental atlas of cortical HFO rate

The regression models, applied to all 114 patients, did not demonstrate any significant developmental growth or diminution of occipital HFO$_{\geq 80\ Hz}$ as defined by any detector (uncorrected $p$-value > 0.05; Fig. 2; Supplementary Fig. 4). In contrast, the rate of extra-occipital HFO$_{\geq 80\ Hz}$ generally showed significant developmental diminution (uncorrected $p$-value < 0.001). Such developmental diminution of extra-occipital HFO resulted in a relatively maintained HFO occurrence rate in the occipital lobes, particularly those adjacent to the calcarine cortex, during adolescence and beyond (Fig. 3B). Below and in Table 3, we describe the findings of HFO$_{HIL \geq 80\ Hz}$ in detail.

Supplementary Movie 3 provides a comprehensive summary of normative development of HFO$_{HIL \geq 80\ Hz}$ rate. Occipital HFO$_{HIL \geq 80\ Hz}$ rate failed to show significant developmental change (uncorrected $p$-value > 0.05; Fig. 2b). In contrast, extra-occipital HFO$_{HIL \geq 80\ Hz}$ rate showed significant developmental diminution (uncorrected $p$-value <

### Table 2 | The results of regression analysis to assess the effect of √age on MI$_{\geq 80\ Hz\ \&\ 0.5–1\ Hz}$

| Lobe | Slope (/√year) | Uncorrected two-sided $p$-value | $t$-value | DF | Lower 95% CI | Upper 95% CI |
|---|---|---|---|---|---|---|
| Frontal | $7.0 \times 10^{-3}$ | *$3.0 \times 10^{-6}$ | 4.7 | 2978 | $4.1 \times 10^{-3}$ | $9.9 \times 10^{-3}$ |
| Temporal | $7.2 \times 10^{-3}$ | *$6.5 \times 10^{-8}$ | 5.4 | 2393 | $4.6 \times 10^{-3}$ | $9.7 \times 10^{-3}$ |
| Parietal | 0.011 | *$2.4 \times 10^{-9}$ | 6.0 | 1999 | $7.4 \times 10^{-3}$ | 0.015 |
| Occipital | 0.047 | *$3.1 \times 10^{-12}$ | 7.1 | 873 | 0.034 | 0.060 |

MI$_{\geq 80\ Hz\ \&\ 0.5\text{-}1\ Hz}$ denotes the strength of coupling between the amplitude of high-frequency oscillation ≥ 80 Hz and the phase of slow-wave$_{0.5\text{-}1Hz}$. $CI$ Confidence interval, $DF$ Degree of freedom. *Significant with False Discovery Rate (FDR) correction on the regression analysis. Supplementary Table 2 presents the results of the analysis, excluding three patients of 21 years old and above.

0.001; Fig. 2b). The assessment of AIC suggested that the developmental diminution in extra-occipital HFO$_{HIL \geq 80\ Hz}$ was comparable during young and older childhood (AIC in extra-occipital lobes: $2.60 \times 10^4$ in age-incorporated model; $2.59 \times 10^4$ in √age model; $2.59 \times 10^4$ in log$_{10}$ age model). The regression slope was −0.41/√year in the frontal lobe, −0.30 /√year in the temporal lobe, −0.33/√year in the parietal lobe, and +0.081 /√year in the occipital lobe (see the detailed statistical results in Table 3). The overall findings on HFO$_{HIL \geq 80\ Hz}$ mentioned above remained qualitatively similar when excluding three patients over 21 years old from the analyses (Supplementary Table 3).

Due to the infrequent occurrence of events, regression models failed to fit the data of HFO$_{\geq 150\ Hz}$ rates in a meaningful manner.

## MI enhancement in the occipital lobe during young childhood and after

The mixed model analysis of iEEG data of 14 children aged from 1.0 and 3.9 suggested that, as compared to the other lobes, the occipital lobe showed higher MI$_{\geq 80\ Hz\ \&\ 0.5\text{-}1\ Hz}$ (mixed model effect: +0.087 [95%CI: +0.078 to +0.097]; uncorrected $p$-value: < 0.001) and MI$_{\geq 80\ Hz\ \&\ 3\text{-}4\ Hz}$ (mixed model effect: +0.033 [95%CI: +0.028 to +0.038]; uncorrected $p$-value: < 0.001).

An ancillary mixed model analysis of iEEG data of 100 individuals aged from 4.0 and 41.5 suggested that, as compared to the other lobes, the occipital lobe likewise showed higher MI$_{\geq 80\ Hz\ \&\ 0.5\text{-}1\ Hz}$ (mixed model effect: +0.17 [95%CI: +0.17 to +0.18]; uncorrected $p$-value: < 0.001) and MI$_{\geq 80\ Hz\ \&\ 3\text{-}4\ Hz}$ (mixed model effect: +0.063 [95%CI: +0.061 to +0.066]; uncorrected $p$-value: < 0.001). These findings indicate that the nonepileptic occipital lobe exhibits higher MI in comparison to the other regions, across all age groups (Fig. 3A). Moreover, the difference in MI between the occipital and extra-occipital lobe regions was found to be approximately two times greater in older individuals.

## Topographical variations of HFO rate during young childhood and after

The mixed model analysis of iEEG data from 14 children between the ages of 1.0 and 3.9 did not show a higher HFO rate in the occipital lobe compared to extra-occipital lobes, except when considering the HFO rate defined by the Hilbert method (HFO$_{STE \geq 80\ Hz}$ mixed model

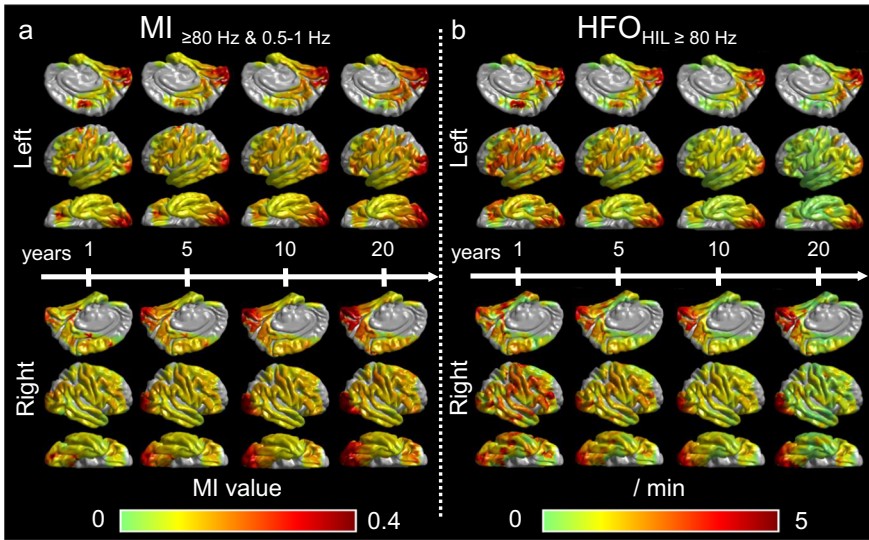

**Fig. 3 | Snapshots of the developmental atlases of cortical MI and HFO. a** Model-predicted MI$_{\geq 80\ Hz\ \&\ 0.5\text{-}1\ Hz}$ and (**b**) HFO$_{HIL \geq 80\ Hz}$ occurrence rate. Each snapshot presents model-predicted values at ages 1, 5, 10, and 20 years. MI Modulation index. MI$_{\geq 80\ Hz\ \&\ 0.5\text{-}1\ Hz}$ denotes the strength of coupling between the amplitude of HFO$_{\geq 80\ Hz}$ and the phase of slow-wave$_{0.5–1Hz}$. The brain images in this figure were created using FreeSurfer (https://surfer.nmr.mgh.harvard.edu/fswiki/CorticalParcellation). HFO$_{HIL}$ High-frequency oscillation defined by the Hilbert method. Supplementary Movies 1–3 present the longitudinal values spanning from 1 to 21 years of age across multiple generations.

**Table 3 | The results of regression analysis to assess the effect of √age on HFO$_{HIL \geq 80 Hz}$ occurrence rate**

| Lobe | Slope (/√year) | Uncorrected two-sided $p$-value | $t$-value | DF | Lower 95% CI | Upper 95% CI |
|---|---|---|---|---|---|---|
| Frontal | −0.41 | *4.3 × 10$^{-54}$ | −15.8 | 2978 | −0.46 | −0.36 |
| Temporal | −0.30 | *7.4 × 10$^{-28}$ | −11.0 | 2393 | −0.36 | −0.25 |
| Parietal | −0.33 | *3.7 × 10$^{-21}$ | −9.5 | 1999 | −0.40 | −0.26 |
| Occipital | 0.081 | 0.31 | 1.0 | 873 | −0.075 | 0.24 |

HFO$_{HIL \geq 80 Hz}$ High-frequency oscillation$_{\geq 80 Hz}$ defined by the Hilbert method, *CI* Confidence interval, *DF* Degree of freedom. *Significant with False Discovery Rate (FDR) correction on the regression analysis. Supplementary Table 3 presents the results of the analysis, excluding three patients of 21 years old and above.

effect: +0.022 [95%CI: −0.039 to +0.082]; uncorrected $p$-value: 0.48; HFO$_{SLL \geq 80 Hz}$ mixed model effect: +0.036 [95%CI: −0.46 to +0.53]; uncorrected p-value: 0.89; HFO$_{HIL \geq 80 Hz}$ mixed model effect: +0.84 [95%CI: +0.57 to +1.1]; uncorrected $p$-value: < 0.001; HFO$_{MNI \geq 80 Hz}$ mixed model effect: −0.044 [95%CI: −0.90 to +3.0 × 10$^{-3}$]; uncorrected $p$-value: 0.067). The model suggests that the nonepileptic occipital lobe of children under four years old generates Hilbert method-defined HFO$_{\geq 80 Hz}$ occurrences at a rate of 0.84/min more frequently than extra-occipital lobe regions.

Another mixed model analysis of iEEG data from 100 individuals of 4.0 years old and above demonstrated that the occipital lobe had a higher HFO rate compared to the extra-occipital lobes, except when considering the HFO rate defined by the MNI method (HFO$_{STE \geq 80 Hz}$ mixed model effect: +0.14 [95%CI: +0.12 to +0.16]; uncorrected $p$-value: < 0.001; HFO$_{SLL \geq 80 Hz}$ mixed model effect: +0.59 [95%CI: +0.43 to +0.75]; uncorrected $p$-value: < 0.001; HFO$_{HIL \geq 80 Hz}$ mixed model effect: +1.8 [95%CI: +1.7 to +1.9]; uncorrected $p$-value: < 0.001; HFO$_{MNI \geq 80 Hz}$ mixed model effect: +0.043 [95%CI: −0.014 to +0.099]; uncorrected $p$-value: 0.14). The results suggest that the nonepileptic occipital lobe of individuals of four years old and above generates Hilbert method-defined HFO$_{\geq 80 Hz}$ occurrences at a rate of 1.8/min more frequently than extra-occipital lobe regions (Fig. 3b).

**Independent effects of development on cortical MI in each lobe**
The mixed model analysis, employed to all 114 patients, confirmed that an increase in √age was associated with an increase in occipital MI$_{\geq 80 Hz \& 0.5-1 Hz}$ (mixed model effect: +0.046 [95%CI: +0.019 to +0.072]; uncorrected $p$-value: 7.43 × 10$^{-4}$; t-value: +3.39; DF: 868) and MI$_{\geq 80 Hz \& 3-4 Hz}$ (mixed model effect: +0.019 [95%CI: +0.0090 to +0.029]; uncorrected $p$-value: 1.22 × 10$^{-4}$; t-value: +3.86; DF: 868), independent of patient and epilepsy profiles. In contrast, mixed model analysis failed to confirm that an increase in √age was associated with an increase in extra-occipital MI$_{\geq 80 Hz \& 0.5-1 Hz}$ or MI$_{\geq 80 Hz \& 3-4 Hz}$ (uncorrected $p$-value: > 0.05). Supplementary Tables 4−7 provide the summarized results of the mixed model analyses to assess the effect of √age on MI$_{\geq 80 Hz \& 0.5-1 Hz}$ in a given lobe. The overall findings remained qualitatively similar when excluding three patients over 21 years old from the analyses (Supplementary Tables 8−11). For interested readers, we have provided Supplementary Tables 12−33, each of which shows the result of mixed model analysis employed to a given region of interest (ROI; Supplementary Fig. 5) instead of a lobe.

**Independent effects of development on cortical HFO rate in each lobe**
The mixed model analysis confirmed that an increase in √age was independently associated with a diminution of HFO$_{HIL \geq 80 Hz}$ rate in the frontal (mixed model effect: −0.53 [95%CI: −0.73 to −0.32]; uncorrected $p$-value: 5.68 × 10$^{-7}$; t-value: −4.97; DF: 2973), temporal (mixed model effect: −0.34 [95%CI: −0.55 to −0.12]; uncorrected $p$-value: 2.0 × 10$^{-3}$; t-value: −3.09; DF: 2388), and parietal lobes (mixed model

effect: −0.38 [95%CI: −0.59 to −0.16]; uncorrected $p$-value: 5.43 × 10$^{-4}$; t-value: −3.46; DF: 1994). However, this phenomenon was not observed in the occipital lobe (mixed model effect: +0.069 [95%CI: −0.27 to +0.41]; uncorrected $p$-value: 0.69; t-value: +0.40; DF: 868). Supplementary Tables 34−37 provide the summarized results of the mixed model analyses to assess the effect of √age on HFO$_{HIL \geq 80 Hz}$ rate in a given lobe. The overall findings remained qualitatively similar when excluding three patients over 21 years old from the analyses (Supplementary Tables 38−41).

**Developmental changes of the spectral frequency band of slow waves nesting HFO**
Looking at MI$_{\geq 80 Hz \& varying slow waves}$, we failed to find a significant developmental change of the frequency band of slow waves nesting HFO (Fig. 4). On average, the regression slope of MI$_{\geq 80 Hz \& s Hz}$ as a function of slow waves (s Hz) was −0.014/Hz in the frontal lobe, −0.013/Hz in the parietal lobe, −0.010/Hz in the temporal lobe, and −0.027/Hz in the occipital lobe, regardless of including or excluding three patients over 21 years old from the analyses. In other words, MI$_{\geq 80 Hz}$ values were generally higher when using lower frequency slow waves. The mixed model analysis failed to demonstrate a significant correlation between this regression slope and patient age in any lobe (uncorrected $p$-value: > 0.05). Thus, the present study failed to demonstrate that the spectral frequency band of slow waves nesting HFO would change with development.

**White matter tracts connecting cortices with developmental MI co-growth**
Univariate regression models, incorporating √age as an independent variable, found that 159 out of the 9464 cortical mesh points (1.7%) showed significantly positive regression slopes, whereas no cortical mesh points showed significantly negative regression slopes. DWI analysis revealed that 2 tracts including the vertical occipital fasciculi and posterior callosal fibers directly connected mesh point pairs showing significant developmental co-growth of MI$_{\geq 80 Hz \& 0.5-1 Hz}$ (Fig. 5 and Supplementary Movie 4).

**White matter tracts connecting cortices with developmental HFO co-diminution**
Univariate regression models incorporating √age as an independent variable found that 902 out of the 9464 cortical mesh points (9.5%) showed significantly negative regression slopes, whereas no cortical mesh points showed significantly positive regression slopes. DWI analysis revealed that 636 tracts, including the arcuate fasciculus, corpus callosum, extreme capsule, frontal aslant tract, inferior fronto-occipital fasciculus, inferior longitudinal fasciculus, middle longitudinal fasciculus, and superior longitudinal fasciculus, directly connected mesh point pairs showing significant developmental co-diminution of HFO$_{HIL \geq 80 Hz}$ rate (Fig. 5 and Supplementary Movie 5).

**Statistical deviation of MI and HFO in the seizure onset zone (SOZ)**
In Supplementary Movies 6 and 7, we have illustrated the normative mean plus two standard deviations of MI biomarkers, as indicated by the regression model, to help readers understand the typical ranges of these markers for given age groups. Furthermore, mixed model analysis employed to each of the four brain lobes demonstrated that z-score normalized MI, reflecting the statistical deviation of MI from the normative mean at given sites and ages, was significantly higher in the SOZ compared to nonepileptic sites. The aforementioned statistical statement was applicable to each of the following MI biomarkers: MI$_{\geq 80 Hz \& 0.5-1 Hz}$, MI$_{\geq 80 Hz \& 3-4 Hz}$, MI$_{\geq 150 Hz \& 0.5-1 Hz}$, and MI$_{\geq 150 Hz \& 3-4 Hz}$ (Table 4). The exception was that MI$_{\geq 150 Hz \& 0.5-1 Hz}$ did not differ between the SOZ and nonepileptic sites within the occipital lobe.

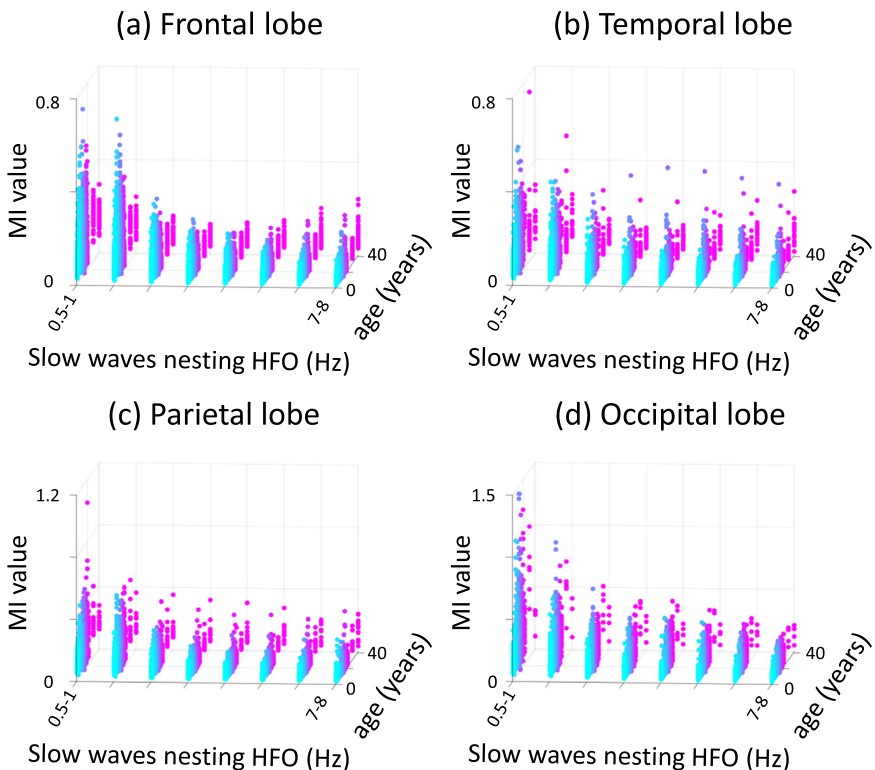

**Fig. 4 | Effect of slow-wave spectral frequency bands on modulation index (MI).** **a** Frontal lobe. **b** Temporal lobe. **c** Parietal lobe. **d** Occipital lobe. $MI_{\geq 80 \text{ Hz \& } s \text{ Hz}}$ denotes the strength of coupling between the amplitude of $HFO_{\geq 80 \text{ Hz}}$ and the phase of slow-wave$_{s \text{ Hz}}$. $MI_{\geq 80 \text{ Hz \& } s \text{ Hz}}$ at a given electrode site is plotted as a function of $s$ being 0.5-1, 1-2, 2-3, 3-4, 4-5, 5-6, 6-7, or 7-8 Hz. Light blue dots: younger individuals. Magenta dots: older individuals. HFO High-frequency oscillation.

Supplementary Movie 8 indicates the mean plus two standard deviations of $HFO_{HIL \geq 80 \text{ Hz}}$ rate for given age groups. Mixed model analysis indicated that z-score normalized $HFO_{HIL \geq 80 \text{ Hz}}$ was higher in the SOZ compared to nonepileptic sites (Table 4). Due to the infrequent occurrence of events, we were unable to compute the statistical deviation of $HFO_{HIL \geq 150 \text{ Hz}}$ in a meaningful manner.

### Relationship between MI and neuropsychological score

A total of 50 patients underwent Peabody Picture Vocabulary Test (PPVT)[36] prior to the surgery; 48 out of the 50 patients had iEEG sampling from the occipital lobe, so they were included in the following statistical analysis. Mixed model analysis failed to show a significant association between z-score normalized $MI_{\geq 80 \text{ Hz \& } 0.5\text{-}1 \text{ Hz}}$ and PPVT score (mixed model effect: $2.4 \times 10^{-9}$; uncorrected $p$-value: >0.99; t-value: $4.4 \times 10^{-5}$; DF: 453).

### Discussion

We have created a developmental atlas of MI, covering the medial and inferior surfaces of the cerebral cortex. It demonstrated physiologically higher MI in the nonepileptic occipital lobe of children younger than 4 years and in the older patient group. The occipital lobe showed higher $MI_{\geq 80 \text{ Hz \& } 3\text{-}4 \text{ Hz}}$ than the other lobes by 0.033 on average in children younger than 4 years and by 0.063 on average in older patients. Although $MI_{\geq 80 \text{ Hz \& } 3\text{-}4 \text{ Hz}}$ is a surrogate marker of interictal spike-and-wave discharges[5,6], the occurrence of physiological HFO nested in slow-wave background activity[11,18,19] inflates $MI_{\geq 80 \text{ Hz \& } 3\text{-}4 \text{ Hz}}$ in the nonepileptic occipital lobe. Therefore, iEEG investigators should cautiously interpret the significance of high-value, occipital MI in presurgical evaluation for young and older patients with drug-resistant focal epilepsy. Nonepileptic occipital lobe sites, especially those proximal to the calcarine sulcus, are expected to show MI values higher than nonepileptic extra-occipital lobe sites (Supplementary Movie 1 and Fig. 3a). With awareness of such topographic variations of MI, iEEG

investigators can reduce the risk of incorrect localization of the epileptogenic zone.

Investigators previously reported the utility of normative atlases of MI, HFO, and other spectral frequency bands, in the presurgical evaluation of patients with the mean age ranging between adolescence and adulthood[7–10,13,37]. In the present study, we have provided atlases presenting the expected mean plus two standard deviations of MI biomarkers to illustrate the typical ranges for given age groups (Supplementary Movies 6, 7). We believe our atlas has the potential to serve as a valuable reference in presurgical evaluations, as z-score normalized MI values were significantly higher in the SOZ compared to nonepileptic sites (Table 4). Compared to $MI_{\geq 80 \text{ Hz \& } 0.5\text{-}1 \text{ Hz}}$, $MI_{\geq 80 \text{ Hz \& } 3\text{-}4 \text{ Hz}}$ showed much larger effect sizes of difference between the SOZ and nonepileptic sites, as suggested by the mixed model estimate (Table 4). This observation can be attributed to the notion that interictal epileptiform discharges are generally associated with a transient increase in HFO stereotypically coupled with a delta wave at 3–4 Hz[6], whereas physiological HFO cycles between augmentation and attenuation during slow-wave sleep at < 1 Hz[18–20]. A prospective study is warranted to investigate whether a more inclusive resection of cortical sites, whose $MI_{\geq 80 \text{ Hz \& } 3\text{-}4 \text{ Hz}}$ values deviate from the age-specific normal range, would predict better postoperative seizure control in young children.

The current study investigated the developmental changes of HFO occurrence rate defined by four open-source detectors (Fig. 2; Supplementary Fig. 4). Commonly across detectors, the HFO rate diminished in the frontal, parietal, and temporal lobes with development. Thus, clinicians need to be aware that young children may show nonepileptic HFO in the extra-occipital lobe regions more frequently than adults. As best demonstrated in Supplementary Movie 3, $HFO_{HIL \geq 80 \text{ Hz}}$ rate was physiologically increased in the occipital regions proximal to the calcarine cortex, but such occipital HFO enhancement was evident during adolescence and adulthood and not during infancy

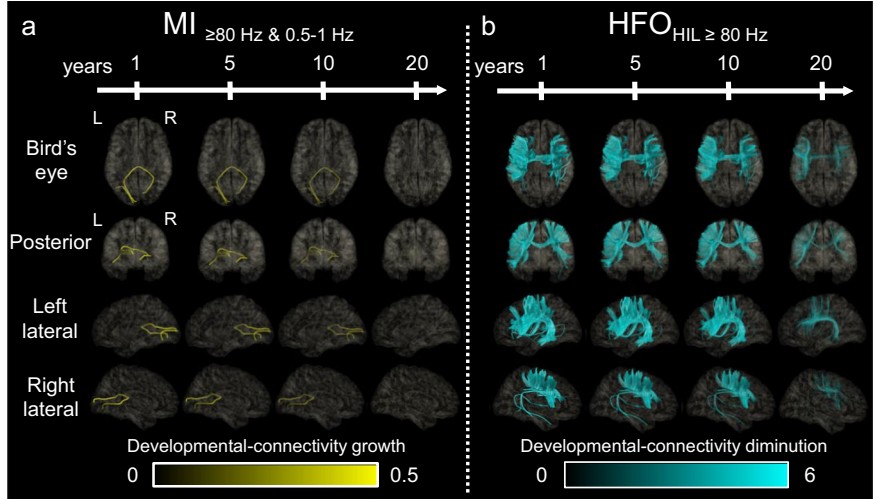

**Fig. 5 | Dynamic tractography.** The video snapshots present the varying intensity of (**a**) co-growth of $MI_{\geq 80\,Hz\,\&\,0.5\text{-}1\,Hz}$ and (**b**) co-diminution of $HFO_{HIL\geq 80\,Hz}$ at ages 1, 5, 10, and 20 years, as estimated by univariate regression analysis incorporating $\sqrt{age}$ as an independent variable. Supplementary Movies 4–5 show the data across generations from 1 to 21 years. The brain images in this figure were created using FreeSurfer (https://surfer.nmr.mgh.harvard.edu/fswiki/CorticalParcellation). MI modulation index. $MI_{\geq 80\,Hz\,\&\,0.5\text{-}1\,Hz}$ denotes the strength of coupling between the amplitude of $HFO_{\geq 80\,Hz}$ and the phase of slow-wave$_{0.5\text{-}1\,Hz}$. $HFO_{HIL}$ High-frequency oscillation defined by the Hilbert method.

**Table 4 | Comparison of intracranial EEG biomarker values between the seizure onset and nonepileptic sites**

| Biomarker | Lobe | Mixed model estimate | SE | t-value | DF | Uncorrected two-sided p-value | Lower 95% CI | Upper 95% CI |
|---|---|---|---|---|---|---|---|---|
| $MI_{\geq 80\,Hz\,\&\,0.5\text{-}1\,Hz}$ | Frontal | 0.75 | 0.070 | 10.8 | 3238 | $1.0 \times 10^{-26}$ | 0.61 | 0.89 |
| | Temporal | 3.00 | 0.48 | 6.25 | 2811 | $4.9 \times 10^{-10}$ | 2.06 | 3.94 |
| | Parietal | 1.27 | 0.11 | 11.8 | 2242 | $4.5 \times 10^{-31}$ | 1.06 | 1.48 |
| | Occipital | 0.45 | 0.14 | 3.17 | 958 | 0.0016 | 0.17 | 0.73 |
| $MI_{\geq 80\,Hz\,\&\,3\text{-}4\,Hz}$ | Frontal | 3.52 | 0.18 | 19.7 | 3238 | $5.0 \times 10^{-82}$ | 3.17 | 3.87 |
| | Temporal | 4.10 | 0.33 | 12.4 | 2811 | $2.0 \times 10^{-34}$ | 3.45 | 4.75 |
| | Parietal | 5.82 | 0.30 | 19.1 | 2242 | $5.8 \times 10^{-75}$ | 5.22 | 6.42 |
| | Occipital | 1.83 | 0.21 | 8.73 | 958 | $1.1 \times 10^{-17}$ | 1.42 | 2.24 |
| $MI_{\geq 150\,Hz\,\&\,0.5\text{-}1\,Hz}$ | Frontal | 0.51 | 0.055 | 9.30 | 3238 | $2.4 \times 10^{-20}$ | 0.40 | 0.62 |
| | Temporal | 2.56 | 0.81 | 3.14 | 2811 | 0.0017 | 0.96 | 4.15 |
| | Parietal | 0.62 | 0.065 | 9.50 | 2242 | $5.3 \times 10^{-21}$ | 0.49 | 0.75 |
| | Occipital | 0.079 | 0.12 | 0.67 | 958 | 0.50 | -0.15 | 0.31 |
| $MI_{\geq 150\,Hz\,\&\,3\text{-}4\,Hz}$ | Frontal | 3.70 | 0.30 | 12.5 | 3238 | $4.7 \times 10^{-35}$ | 3.12 | 4.28 |
| | Temporal | 3.77 | 0.73 | 5.16 | 2811 | $2.6 \times 10^{-7}$ | 2.34 | 5.20 |
| | Parietal | 5.41 | 0.43 | 12.5 | 2242 | $1.3 \times 10^{-34}$ | 4.56 | 6.26 |
| | Occipital | 0.94 | 0.18 | 5.11 | 958 | $3.9 \times 10^{-7}$ | 0.58 | 1.31 |
| $HFO_{HIL\geq 80\,Hz}$ | Frontal | 2.45 | 0.079 | 30.9 | 3238 | $2.4 \times 10^{-184}$ | 2.30 | 2.61 |
| | Temporal | 1.85 | 0.080 | 23.3 | 2811 | $8.5 \times 10^{-110}$ | 1.70 | 2.01 |
| | Parietal | 2.95 | 0.10 | 28.7 | 2242 | $1.0 \times 10^{-154}$ | 2.75 | 3.16 |
| | Occipital | 1.59 | 0.14 | 11.4 | 958 | $1.6 \times 10^{-28}$ | 1.32 | 1.86 |

The results of mixed model analyses are provided. All z-score normalized biomarker values mentioned in this table (except $MI_{\geq 150\,Hz\,\&\,0.5\text{-}1\,Hz}$) were significantly higher in the seizure onset than in the nonepileptic sites. *CI* 95% confidence interval. *DF* degree of freedom. *SE* standard error. $MI_{\geq f\,Hz\,\&\,s\,Hz}$ denotes the strength of coupling between the amplitude of $HFO_{\geq f\,Hz}$ and the phase of slow-wave$_{s\,Hz}$. $HFO_{HIL\geq 80\,Hz}$ denotes the rate of high-frequency oscillation defined by the Hilbert method. We could not compute z-score normalized $HFO_{HIL\geq 150\,Hz}$ because the regression model failed to fit the data of $HFO_{HIL\geq 150\,Hz}$ rates in a meaningful manner, due to the infrequent occurrence of events.

or toddlerhood. Supplementary Movie 8 presents the mean plus two standard deviations of the $HFO_{HIL\,\geq 80\,Hz}$ rate, as suggested by the regression model. This atlas may be useful in helping investigators understand the typical range of this biomarker value at given sites and given ages. Our mixed model analysis demonstrated that, within each brain lobe, z-score normalized $HFO_{HIL\geq 80\,Hz}$ was higher in the SOZ compared to nonepileptic sites (Table 4).

$HFO_{MNI\,\geq 80\,Hz}$ rate was very low in the occipital lobes throughout all ages; this MNI method-specific observation can be attributed to its detector being designed to be agnostic to persistent forms of high-frequency activity[17], as often seen in the nonepileptic occipital lobe[11,38].

What is responsible for the developmental enhancement of MI in the occipital lobe? The current study found that $MI_{\geq 80\,Hz\,\&\,0.5\text{-}1\,Hz}$ increases rapidly during early childhood and modestly thereafter. However, the occipital HFO rate did not increase significantly with development, suggesting that rate changes may not sufficiently account for the developmental enhancement of MI (Fig. 2b). Furthermore, an assessment of the spectral frequency bands of slow-waves

nesting HFO (Fig. 4) did not provide evidence that such nesting slow rhythms become faster with development. In contrast, our white matter connectivity assessment revealed that occipital regions exhibiting a developmental enhancement of MI were accompanied by monosynaptic white matter streamlines, including the vertical occipital fasciculi and posterior callosal fibers (Fig. 5a). These white matter structures are essential for object recognition[39–42]; thus, one possible explanation for the developmental enhancement of occipital MI is that the cerebral cortex learns to generate HFO at a preferred phase of background rhythms in an experience-dependent manner. This notion is supported by the observation that monocular sight deprivation during a critical period altered delta-gamma phase-amplitude coupling in the mouse visual cortex[43].

In the current study, we found no significant association between the z-score normalized $MI_{\geq 80\,Hz\,\&\,0.5\text{-}1\,Hz}$ in the occipital lobe and the age-normalized PPVT score, an instrument measuring visual object recognition. Given that we failed to find an association between MI and PPVT, one must be cautious in extrapolating a causal relationship between delta-nested HFO and visual memory consolidation. Previous research has employed more targeted methodologies to discern the impact of EEG measures on visual memory consolidation[44–46]. These studies incorporated a pre-sleep visual task, followed by a behavioral test upon awakening—a process that may more effectively reveal the causal relationship between cortical signals and visual memory consolidation. A study of 12 patients through invasive EEG sampling predominantly from the temporal lobe regions reported that gamma activity[30-90 Hz] in the temporal lobe, when replayed during sleep, was associated with successful recall upon awakening[46]. Conversely, our study did not provide any of the participating patients with a pre-sleep visual task or a post-sleep behavioral test. A study with a proper visual perceptual task[45] and behavioral assessment before and after sleep is necessary to better determine the causal relationship between occipital delta-nested HFO and visual memory consolidation.

The methodological innovations in the current study include the generation of developmental atlases based on iEEG signals from 114 patients, including infants and toddlers. We previously validated our group-level analysis of iEEG across generations from infancy to adulthood[47]. A pivotal procedure to ensure the spatial accuracy of electrode-brain surface coregistration was a manual delineation of the pial surface in the temporal lobe regions that are frequently unmyelinated in young children[47].

Visualization of the direct white matter streamlines connecting cortical regions with a developmental enhancement of MI is an application of dynamic tractography technology[33,48]. We interpret the visualized streamlines, including vertical occipital fasciculi and posterior callosal fibers, to reflect the ongoing development of large-scale neural communications based on slow wave-nested HFO before reaching the adult level.

Inevitable limitations in the present study include limited spatial sampling, though 8251 nonepileptic electrode sites were available for analysis. We are able to place electrodes that are clinically indicated. Neither Heschl nor insular gyri were analyzed in the present study using subdural electrodes. Thus, our study was not designed to assess the developmental enhancement of MI or HFO in much of the primary auditory cortex. We did not sample iEEG signals from the thalamus either because there was no clinical indication. Since stereotactic-EEG depth electrodes are used to assess the thalamic nuclei to determine the optimal target for deep brain stimulation and responsive neurostimulation therapy, further studies are expected to determine the developmental changes of MI and HFO in such deep brain structures in the future.

Another limitation included cross-sectional measurement due to the invasive nature of iEEG recording. Chronic iEEG recording is available through eight intracranial electrode contacts in patients undergoing responsive neurostimulation. Yet, longitudinal assessment of nonepileptic iEEG signals using these contacts may not be feasible because these electrodes are implanted in the area proximal to the epileptogenic zone.

We determined the age effect on iEEG measures by employing an FDR correction taking into account that we repeated the mixed model analysis 18 times for six iEEG measures and three types of age. This analytic approach reduced the risk of Type I error while increasing the risk of Type II error[49]. Thus, the reported absence of statistical difference should be interpreted as failure to find a difference.

Our ancillary analysis on the rates of $HFO_{\geq 150\,Hz}$ revealed that the infrequent occurrence of these events made it impossible for regression models to meaningfully fit the data. To detect sufficient numbers of $HFO_{\geq 150\,Hz}$ events for analysis of developmental changes, interictal slow-wave sleep epochs much longer than 20 min would be necessary. However, the duration of iEEG recording is determined solely by clinical needs, making it unrealistic to expect that all study participants would have prolonged iEEG epochs that meet the eligibility criteria employed. Although the spatial distribution of the rates of $HFO_{\geq 80\,Hz}$ and $HFO_{\geq 150\,Hz}$ are generally similar to each other, some studies suggest that $HFO_{\geq 250\,Hz}$ may be more specific than $HFO_{\geq 80\,Hz}$ in localizing the epileptogenic zone; however, others imply that $HFO_{\geq 250\,Hz}$ may not be as practical as $HFO_{\geq 80\,Hz}$ due to its rarity and its presence in the nonepileptic eloquent cortex[1,2,7,50].

One drawback of using iEEG is that we are unable to include healthy controls, because it is unethical to implant subdural electrodes without clinical indication. Thus, we excluded electrode sites affected by either SOZ, interictal spike zone, and MRI-visible lesions from the analysis. Furthermore, we employed mixed model analysis to control the effects of potential confounding factors on MI/HFO measures, including antiseizure medications, presence of MRI-visible lesions, and locations of SOZ. Despite those issues, iEEG involves benefits over scalp recording such as, >100 times better signal fidelity and sampling from the medial or inferior cerebral cortex surface[51].

We did not ask any of the patients about dream contents immediately following arousal; therefore, we are unable to determine whether developmental changes in MI and HFO are attributed to the change in dreaming behaviors during slow-wave sleep[52].

In summary, our study of 114 patients achieving ILAE class-1 outcome demonstrated that z-score normalized MI and $HFO_{\geq HIL80\,Hz}$ rate were significantly higher within the SOZ compared to nonepileptic sites in each lobe. A prospective study is warranted to investigate whether a more inclusive resection of cortical sites, whose iEEG biomarker values deviate from the age-specific normal range, would predict better postoperative seizure control in young children.

## Methods
### Patients
The inclusion criteria consisted of [a] simultaneous video-iEEG recording between January 2007 and November 2020, as part of pre-surgical assessment at Children's Hospital of Michigan or Harper University Hospital, Detroit Medical Center, [b] iEEG sampling rate of 1000 Hz[53], [c] iEEG contained an artifact-free 20 min slow-wave sleep epoch at least two hours apart from clinical seizure events[11,54], and [d] International League Against Epilepsy (ILAE) class 1 outcome in the last follow-up after focal resection[7]. The exclusion criteria included [a] a history of previous resective epilepsy surgery, [b] undergoing hemispherotomy or hemispherectomy, [c] lacking artifact-free, nonepileptic electrode sites (defined as those outside the SOZ[55], interictal spike zone[6], and MRI-visible lesions[56]), and [d] the presence of massive brain malformations (such as megalencephaly and perisylvian polymicrogyria) that make it difficult to reliably identify the central, lateral, or calcarine sulci[57]. The Institutional Review Board of Wayne State University approved the current study, and we obtained written informed consent from the patients' legal guardians and written assent from pediatric patients aged 13 years or older.

### Extraoperative video-iEEG

We acquired extraoperative video-iEEG data using the same protocols as previously reported[7,11,47]. A licensed neurosurgeon surgically implanted platinum subdural disk electrodes (3 mm diameter and 10 mm center-to-center distance) on the hemisphere suspected to contain the epileptogenic zone, followed by the placement of surface electromyographic electrodes on the deltoid muscles and electro-oculographic electrodes (2.5 cm below and 2.5 cm lateral to the outer canthi) to assess body movements during iEEG recording[11]. Intracranial EEG recording aimed to determine the boundaries of the presumed epileptogenic zone and functionally important cortices. The spatial extent and duration of iEEG sampling were solely determined by clinical needs specific to each patient. Following implantation, iEEG data was continuously recorded at the bedside with a sampling rate of 1000 Hz, for 2–7 days. To ensure the accuracy of the developmental iEEG atlases, we included artifact-free iEEG segments from none-pileptic electrode sites alone.

### MRI

We obtained preoperative 3-tesla MRI data, including T1-weighted spoiled gradient-echo volumetric and fluid-attenuated inversion recovery images[47,57]. The FreeSurfer software package was used to reconstruct the MRI surface image of patients aged two and above (http://surfer.nmr.mgh.harvard.edu)[58,59]. In cases where the software failed to detect the pial surface accurately due to insufficient cerebral myelination, a board-certified neurosurgeon (K.S.) manually delineated the pial surface using the Control Point function (https://surfer.nmr.mgh.harvard.edu/fswiki/FsTutorial/ControlPoints_freeview/)[47,60–62]. For patients younger than two, we used the Infant FreeSurfer software package to reconstruct the surface image (https://surfer.nmr.mgh.harvard.edu/fswiki/infantFS)[47,63].

We displayed electrode sites on the pial surface of the brain using preoperative MRI and a post-implant CT image[47,64]. Two board-certified neurosurgeons (K.S. and N.K.) visually assessed intraoperative photographs to confirm the spatial accuracy of electrode locations co-registered to the MRI surface image[65]. In order to pool sites from all patients, it was necessary to normalize the electrode locations to the standardized FSaverage brain surface (http://surfer.nmr.mgh.harvard.edu). In a previous iEEG study of 32 patients, we found that the mean coregistration error ranged below 0.4 mm, and there was no significant correlation between patient age and the severity of coregistration error[47].

### Focal resection and postoperative seizure outcome

In our previous studies[55,66], we described the guiding principle for determining the boundary of cortical resection. Our aim was to remove the SOZ and any adjacent MRI lesions, while preserving functionally-important cortex; this procedure is intended to maximize seizure control and minimize development of cognitive and/or sensorimotor deficits. Importantly, none of the data from this study was available to inform clinical decision making. Consistent with the study design, all 114 patients achieved ILAE Class-1 outcome at the last follow-up, which occurred at least one year after surgery.

### Identification of slow-wave sleep

We visually identified and analyzed the earliest available, artifact-free, 20 min slow-wave sleep iEEG epochs that were at least 2 ho apart from ictal events[54,67]. We ensured that slow-wave sleep iEEG signals at non-epileptic sites displayed low-frequency ( < 2 Hz) activity for at least 20% of the time during each 30 s epoch. We used Rayleigh's test to examine if the distribution of the onset time of the analyzed slow-wave sleep epochs deviated from a uniform distribution. Furthermore, we employed Spearman's rank test to investigate any potential correlation between patient age and the onset time of the analyzed slow-wave sleep epoch.

### Modulation index (MI)

At each artifact-free nonepileptic site, we used the EEGLAB winPACT toolbox (https://sccn.ucsd.edu/wiki/WinPACT)[4] to automatically compute MI during a 20 min slow-wave sleep epoch. We employed the EEG.etc.winPACT.canoltysMIAllChan command within the winPACT toolbox (Supplementary Figure 6; https://github.com/sccn/winPACT). This program transforms all iEEG data points into Hilbert spectra and quantifies the strength of coupling between HFO amplitude and the instantaneous phase of slow waves. In the notation $MI_{\geq f\ Hz\ \&\ s\ Hz}$ denotes the strength of coupling between the amplitude of $HFO_{\geq f\ Hz}$ and the phase of slow-wave$_{s\ Hz}$.

We report the normative developmental changes primarily in $MI_{\geq 80\ Hz\ \&\ 0.5\text{-}1\ Hz}$ (Fig. 2), a surrogate marker of the delta-nested HFO that have been hypothesized to underly consolidation of long-term memory related to given sensory representations[11,23–26]. We also report the normative developmental changes in $MI_{\geq 80\ Hz\ \&\ 3\text{-}4\ Hz}$, which are suggested to be increased in iEEG traces showing frequent interictal spike-and-wave discharges[5,6]. The normative spatial variability of this biomarker value could be a valuable reference in epilepsy presurgical evaluation.

### High-frequency oscillation (HFO)

During the same 20 min slow-wave sleep iEEG epochs described above, we used the RIPPLELAB software (https://github.com/BSP-Uniandes/RIPPLELAB/)[68] to compute the HFO rate at each artifact-free none-pileptic site, using four different detection algorithms: [1] Short Time Energy (STE) method[14], [2] Short Line Length (SLL) method[15], [3] Hilbert (HIL) method[16], and [4] Montreal Neurological Institute (MNI) method[17]. The STE method defines a $HFO_{STE\ \geq f\ Hz}$ event as an iEEG segment presenting successive root mean square values greater than five standard deviations above the overall mean of root mean squares and containing more than six peaks greater than three standard deviations on the $f$-Hz high-pass filtered iEEG trace. The SLL method defines a $HFO_{SLL\ \geq f\ Hz}$ as an iEEG segment presenting SLL amplitude augmentation greater than the 97.5th percentile of the empirical cumulative distribution function computed on the $f$-Hz high-pass filtered iEEG trace. The HIL method defines a $HFO_{HIL\ \geq f\ Hz}$ as an iEEG segment presenting Hilbert transform-based envelope augmentation greater than five standard deviations on the $f$-Hz high-pass filtered iEEG trace. The MNI method defines a $HFO_{MNI\ \geq f\ Hz}$ as an iEEG segment presenting root mean square energy above the 99.9999th percentile compared to the baseline. If the baseline was absent due to persistent high-frequency activity, the MNI method treated an iEEG segment presenting the ≥95th percentile of the cumulative distribution function as computed on the 1 min $f$-Hz high-pass filtered iEEG trace. We used the default parameter settings in the RIPPLELAB software, as reported previously[7]. In the present study, we focused on reporting the developmental changes in the occurrence rate of $HFO_{HIL\ \geq 80\ Hz}$ (/min) as HFO event was defined commonly on Hilbert transformed iEEG traces when quantifying the $HFO_{HIL}$ occurrence rate and computing MI.

An ancillary analysis was performed on the rates of $HFO_{\geq 150\ Hz}$ for interested readers.

### Statistical analysis to confirm physiological enhancement of MI and HFO in the occipital lobe during young childhood

In a previous study, we created normative atlases of MI and HFO based on data from 47 patients aged between 4 and 19 years, which revealed a general enhancement of these measures in the occipital lobe[7]. In the current study, we aimed to replicate this finding in a cohort of 14 children aged between 1.0 and 3.9 years and 100 individuals aged 4 years or older. We used mixed model analysis with electrode location in the occipital lobe (yes = 1) as the fixed effect predictor variable and either $MI_{\geq 80\ Hz\ \&\ 0.5\text{-}1\ Hz}$, $MI_{\geq 80\ Hz\ \&\ 3\text{-}4\ Hz}$, $HFO_{STE\ \geq 80\ Hz}$, $HFO_{SLL\ \geq 80\ Hz}$, $HFO_{HIL\ \geq 80\ Hz}$, or $HFO_{MNI\ \geq 80\ Hz}$ as the dependent variable. The random

effect factors included intercept and patient. We considered an FDR-corrected p-value of less than 0.05 as significant, for comparisons of six iEEG measures. We reported the mixed model effect and 95% confidence interval (95% CI) to highlight the impact of topography on normative iEEG biomarker measures. All statistical analyses were performed using Matlab R2020a (MathWorks Inc., Natick, MA).

### Statistical analysis to visualize the developmental slope of cortical MI and HFO at given mesh points

We presented iEEG biomarker measures at each nonepileptic electrode site on a standardized brain surface image using FreeSurfer and interpolation within 10 mm from the electrode center at the individual patient level[47]. Using linear and nonlinear univariate regression models at cortical surface mesh points consisting of 20 neighboring Free-Surfer vertex finite elements[47,58], we then determined the developmental slope of the iEEG biomarker at given mesh points. We used age, square root of age ($\sqrt{age}$), and $\log_{10}$ age as an independent variable, and $MI_{\geq 80\ Hz\ \&\ 0.5\text{-}1\ Hz}$, $MI_{\geq 80\ Hz\ \&\ 3\text{-}4\ Hz}$, $HFO_{STE\ \geq 80\ Hz}$, $HFO_{SLL\ \geq 80\ Hz}$, $HFO_{HIL\ \geq 80\ Hz}$, or $HFO_{MNI\ \geq 80\ Hz}$ as dependent variables. We evaluated the goodness of fit of each regression model using AIC. A biomarker value was considered to increase with age if the regression slope was significantly greater than zero. We identified the mesh points where the developmental change (enhancement or diminution) of a given iEEG biomarker measure survived an FDR correction (for 18 comparisons: six iEEG measures × three types of age measures) in the resulting normative atlas. We also created video atlases, each displaying $MI_{\geq 80\ Hz\ \&\ 0.5\text{-}1\ Hz}$, $MI_{\geq 80\ Hz\ \&\ 3\text{-}4\ Hz}$, or $HFO_{HIL\ \geq 80\ Hz}$ values predicted by a given regression model at cortical mesh points (Fig. 3; Supplementary Movies 1–3).

### Statistical analysis to determine the independent effects of development on cortical MI and HFO in each lobe

We used mixed model analysis[33,47] to determine the lobe where a developmental change of a given iEEG biomarker remained significant, after controlling for the independent effects of patient demographics (age and sex) and epilepsy-related variables (SOZ location, MRI lesion, and number of oral antiseizure medications). The aim was to account for potential confounders that could affect MI and HFO measures.

To assess the developmental change of MI at each lobe, we utilized the MATLAB fitlme command (https://www.mathworks.com/help/stats/fitlme.html) to fit a mixed model specified by the following formula: 'MI - 1 + age + sex + SOZ + MRI + hemisphere + number of antiseizure medications + (1|patient)'. Here, the dependent variable was MI at a given analysis mesh point, and the fixed effect predictors included [1] age at surgery (e.g., $\sqrt{year}$), [2] sex (female = 1), [3] presence of SOZ in a given lobe (yes = 1), [4] presence of MRI-visible structural lesion (yes = 1), [5] sampled hemisphere (left = 1), and [6] number of oral antiseizure medications taken immediately before the initiation of iEEG recording. We considered a larger number of antiseizure medications as a surrogate of a more severe seizure-related cognitive burden since polytherapy is associated with more disabling seizures and seizure-related cognitive impairment[7]. We employed this approach since no single neuropsychological assessment can quantify the severity of cognitive impairment across all age ranges. Our random effect factors included the intercept and patient. We deemed an FDR-corrected p-value of 0.05 (for 18 comparisons: six $iEEG_{\geq 80\ Hz}$ measures × three types of age measures) as the significance threshold.

### Statistical analysis to determine if the spectral frequency band of slow waves nesting HFO would change with development

Some might hypothesize that slow waves with a higher spectral frequency band would facilitate a higher rate of spontaneous neural communication through nested HFO. As an ancillary analysis, we, therefore, sought to determine whether the spectral frequency band

of slow waves nesting HFO would change with development. To accomplish this, we measured $MI_{\geq 80\ Hz\ \&\ s\ Hz}$, where s was 0.5-1, 1-2, 2-3, 3-4, 4-5, 5-6, 6-7, or 7-8 (Fig. 4) and calculated the regression slope of $MI_{\geq 80\ Hz\ \&\ s\ Hz}$ with respect to s Hz, in each lobe. We then assessed whether this regression slope was dependent on patient age in each lobe using the Spearman correlation. A higher regression slope in older individuals, compared to younger individuals, would suggest that $HFO_{\geq 80\ Hz}$ in older individuals were preferentially coupled with slow waves of higher spectral frequency bands. We set an FDR-corrected p-value of 0.05 (for testing at four lobes) as the significance threshold.

### Statistical analysis: white matter tracts between cortices showing developmental MI co-growth

We visualized white matter tracts directly connecting cortical mesh points that showed significant developmental co-growth (or co-reduction) of $MI_{\geq 80\ Hz\ \&\ 0.5\text{-}1\ Hz}$. To this end, we used the regression slope of $MI_{\geq 80\ Hz\ \&\ 0.5\text{-}1\ Hz}$ as a function of $\sqrt{age}$ at each cortical mesh point, as computed in a regression analysis mentioned above. We declared that the developmental enhancement (or reduction) of nested HFO-based neural communications took place between two cortical mesh points if [1] two distinct mesh points showed significantly positive (or negative) regression slopes, and [2] these mesh points were accompanied by direct tractography streamlines on diffusion-weighted imaging (DWI) analysis[48].

We delineated white matter DWI streamlines using open-source data from 1065 healthy participants (http://brain.labsolver.org/diffusion-mri-templates/hcp-842-hcp-1021)[69], as previously reported[33,48]. Our group validated the use of open-source DWI data by demonstrating that the inferred velocity of neural propagations induced by single-pulse electrical stimulation was similar whether using open-source or individual patient DWI data[33]. We placed seeds (4 mm radius) at cortical mesh points demonstrating significant developmental enhancement (or reduction) of nested HFO-based neural communications (as rated by $MI_{\geq 80\ Hz\ \&\ 0.5\text{-}1\ Hz}$). Using DSI Studio (http://dsi-studio.labsolver.org/), we visualized DWI streamlines directly connecting the mesh points within Montreal Neurological Institute standard space. For fiber tracking, we utilized the following parameters: a quantitative anisotropy threshold of 0.05, a maximum turning angle of 70°, and a streamline length of 20 to 250 mm. In this investigation, we exclusively visualized DWI streamlines with at least 50% of their coordinates in one of the following white matter tracts: arcuate fasciculus, cingulum, corpus callosum, extreme capsule, frontal aslant tract, inferior fronto-occipital fasciculus, inferior longitudinal fasciculus, middle longitudinal fasciculus, superior longitudinal fasciculus, uncinate fasciculus, or vertical occipital fasciculus, as defined in DSI Studio (as previously performed in Sonoda et al. [33]). We excluded streamlines involving the brainstem, basal ganglia, thalamus, or cerebrospinal fluid space.

The resulting dynamic tractography video atlases highlighted the intensity of $MI_{\geq 80\ Hz\ \&\ 0.5\text{-}1\ Hz}$ developmental co-growth (or co-reduction) via given tractography streamlines for every 0.1 years; thereby, the intensity was defined as ($\sqrt{|\text{regression slope at a mesh point}|}$ × $\sqrt{|\text{regression slope at another mesh point}|}$), at a given streamline connecting a pair of mesh points (Fig. 5).

### MI and HFO in the seizure onset zone (SOZ)

We employed mixed model analyses in given brain lobes to determine whether the SOZ electrode sites exhibited significantly higher MI and HFO rates compared to nonepileptic sites. Here, the dependent variable was z-score normalized $MI_{\geq 80\ Hz\ \&\ 0.5\text{-}1\ Hz}$, $MI_{\geq 80\ Hz\ \&\ 3\text{-}4\ Hz}$, $MI_{\geq 150\ Hz\ \&\ 0.5\text{-}1\ Hz}$, $MI_{\geq 150\ Hz\ \&\ 3\text{-}4\ Hz}$, $HFO_{HIL\geq 80\ Hz}$, and $HFO_{HIL\geq 150\ Hz}$ at a given electrode site. We computed a z-score normalized value of a given iEEG biomarker using the mean and standard deviation across 30 nonepileptic electrode sites closest to a given electrode contact[7]; thereby, for children who are 'n' years old, we computed the mean

and standard deviation across children between 'n' and 'n + 3.9' years old. The fixed effect predictor included the SOZ label (SOZ = 1; nonepileptic = 0). The random effect factors included the intercept and patient. We deemed an FDR-corrected $p$-value of 0.05 (for six comparisons: six iEEG measures) as the significance threshold.

We have illustrated the mean plus two standard deviations of the aforementioned iEEG biomarkers, as indicated by the regression model, to help readers understand the typical ranges of these markers for given age groups (Supplementary Movies 6–8).

### Relationship between MI and neuropsychological score

To explore the relationship between occipital MI and visual object recognition skills, we employed mixed model analysis on the patients who had undergone the PPVT[36] before surgery and had at least one nonepileptic electrode site within the occipital lobe. The dependent variable was the baseline PPVT score, and the fixed effect predictor was z-score normalized $MI_{\geq 80\ Hz\ \&\ 0.5\text{-}1\ Hz}$. The random effect factors included intercept and patient.

### Reporting summary

Further information on research design is available in the Nature Portfolio Reporting Summary linked to this article.

## Data availability

The iEEG data are available at https://openneuro.org/datasets/ds004551/versions/1.0.6 (https://doi.org/10.18112/openneuro.ds004551.v1.0.6)[70]. Source data are provided with this paper.

## Code availability

The analysis codes are available at https://github.com/kaz1126/MI_HFO (https://doi.org/10.5281/zenodo.8267570)[71].

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

## Acknowledgements
We are grateful to Karin Halsey, BS, REEGT, and Jamie MacDougall, RN, BSN, CPN at Children's Hospital of Michigan, for the collaboration and assistance in performing the studies described above. This work was supported by the National Institutes of Health (NS064033 to E.A.) and by Japan Society for the Promotion Science (JP22J23281 to N.K.).

## Author contributions
Study design: K.S., N.K., E.A.; Data acquisition: K.S., N.K., M.S., T.M., A.F.L., N.I.M., S.S., E.A.; Analysis: K.S., N.K., M.S., E.A.; Writing original draft: K.S., E.A.; Writing review and editing: E.F., E.A.; Supervision: E.A.

## Competing interests
The authors declare no competing interests.
