## [Peer Review File · Nature Communications]

Developmental atlas of phase-amplitude coupling between physiologic high-frequency oscillations and slow wavesREVIEWER COMMENTS

Reviewer #1 (Remarks to the Author):

The authors recorded intracranial EEG (iEEG) during the presurgical workup in patients with epilepsy. There are several biomarkers in the iEEG to guide epilepsy surgery. Currently, high frequency oscillations (HFOs) are proposed as a new biomarker as well as their relation to slow waves (MI). The authors studied HFO and MI in brain healthy tissue, far from the supposed epileptogenic zone. One motivation of the study is to provide a normative atlas of background activity against which pathological activity can be identified. A second motivation is to present the development of MI with age during childhood, where MI may be a marker of visual memory consolidation.

The number of patients is fairly large. The methods use standard toolboxes for iEEG time series analysis and image processing. This study extends a previous similar publication from the MNI group (Frauscher et al., Ann Neurol 2018) to patients during their childhood. The data analysis seems solid. The interpretation of MI as indicator for visual memory consolidation is somewhat speculative: The visual memory consolidation was not tested in these patients but only inferred from the literature. The conclusions with respect to epilepsy surgery should be more cautious.

In particular, the manuscript must be improved in several ways.

1. The iEEG figures must be improved. All the values given in the legend should be presented in a table and not in the figure panel.
2. The authors use the same y-axis in all plots for better comparison across brain areas. This puts many points close to $y=0$ so that they cannot be discerned. I suggest a \log_{10} y-scale combined with a \log_{10} x-scale. Then the fit of a power law will appear as a straight line with the slope being the exponent. Given the three linear regressions used in this study (age, square root of age, and \log_{10} age), this will

be more illustrative. At the same time, it will allow a visual judgement of the regression, which is not possible now for many subplots.

3. The three patients >20years appear as outliers. Please redo all analyses without these three patients to verify that you results are valid for your core set of patients, which is aged ≤ 20 years.

4. The authors record iEEG with 1000 Hz. This precludes analysis of spectral content >250Hz. All their text on HFO >250Hz (Fast Ripples) is unfounded speculation not based on their data. This text should simply be removed.

5. Please read the reference (Zweiphenning et al., 2022a) closely and summarize it correctly. The current summary is wrong.

6. Please remove the older reviews on HFO and replace them by (Chen Z, Maturana MI, Burkitt AN, Cook MJ, Grayden DB. High-frequency Oscillations in Epilepsy: What Have We Learned and What Needs to be Addressed. Neurology. 2021. <https://doi.org/10.1212/wnl.00000000000011465>). I recommend reading that review closely. While the study analyses four types of HFO for which detectors are available in the toolbox, their value for epilepsy surgery is still debated.

7. While the conclusions regarding epilepsy appear in the abstract, they should be made explicit at the end of the manuscript. Of course, they must be cautious and limited to the data provided in this study. Explicit conclusions will help the reader to understand the noteworthiness of the study.

8. Please label the video file names for easier access.

Reviewer #2 (Remarks to the Author):

This paper describes the presumed physiological activity (HFOs and MI) in 114 patients with focal epilepsy using invasive EEG electrodes outside the diseased areas. It creates a topographical and age-related 4-dimensional atlas.

I think this is a very nice and relevant attempt, yielding a beautiful data representation useful for researchers in neuroscience from many directions. The 4-dimensionality is novel.

The work supports the conclusions and claims.

I did not notice evident flaws.

The methodology is generally sound and enough detail is provided.

I have two general and several specific comments.

General

1) The writing is generally good, but I believe it can be much more structured (adhering strictly to intro-methods-results sections and use the same structures within). I will mention some examples in my comments below.

2) Sentences can be shorter. E.g., words like 'only' or 'in this study' are often redundant and words like 'as well as' can be shortened to 'and'

Specific

1) Abstract/manuscript: iEEG included only subdural (electrocorticography) recordings. This should be specified, already in the abstract

2) Introduction: 'independent effects of patient and epilepsy profiles': how is epilepsy profiles an independent effect? I could not find the definition of 'epilepsy profiles' in methods, but it is used several times in abstract, introduction and results.

3) Introduction: 'In addition to modeling cortical ontogenic changes, the present study also visualized 140 white matter pathways supporting the development of neural communications via delta141 nested HFO (as rated by $MI \geq 80$ Hz & 0.5-1 Hz), during slow-wave sleep. To accomplish this, we 142 utilized our novel imaging technique, dynamic tractography, which combines iEEG signals 143 with MRI tractography' This is a methods-like description while in the introduction I want to hear why this extra step was taken and why the specific methods.

4) Methods: a sample rate of 1000 Hz is low to recognize HFOs > 250 Hz

5) Methods/results: the methods describe that 114 patients will be selected and then the inclusion criteria. I want to know how these 114 patients resulted from how many patients in the beginning. It seems that the period is the strict criterium, so than 114 patients is a result from the selection.

6) Methods/results: To ensure the accuracy of the developmental iEEG 192 atlases, we only included artifact-free iEEG segments from 8,251 nonepileptic electrode sites 193 (mean: 72.4 per patient; range: 12 to 121;) Here again, the number of used/selected channels is a result. I want to know how many channels were discarded because of artefacts, SOZ, MRI, irritative zones.

7) survived an FDR correction (for 18 comparisons: six iEEG measures \times three types of age measures): I wonder if an FDR is really needed here because both the iEEG measures and the types of age measures are dependant variables. It is good to reduce the change of finding false positive relations but high correction for false discoveries may yield false negative results. This should be discussed in the discussion (Wagenmakers E et al Nature 2022 one statistical analysis must not rule them all).

Reviewer #3 (Remarks to the Author):

I read with interest the paper “Developmental atlas of phase-amplitude coupling between physiologic high-frequency oscillations and slow waves” by Prof Asano and colleagues.

In my opinion, the most important result of this paper is the cognitive aspect of the developing brain and the link with MI (an index that evaluates the phase amplitude coupling of HFO and delta waves). In particular, they found that MI is strengthened during development in the occipital lobe and that the cortical regions that show this co-growth are connected by white matter fibers, namely the vertical occipital fasciculi and the posterior callosal fibers. These results could be explained by the visual memory consolidation in early childhood.

Moreover, this could be of help in the analysis of the epileptogenic biomarkers sensing to delineate the epileptogenic zone network – EZN - (HFOs and MI), by a better understanding of physiological activities during development.

However, I have to two main criticisms to this work:

1. the absence of analysis on the pathological channels and the claim of a better definition of the EZN
2. the lack of evidence supporting the hypothesis that visual memory consolidation and the strengthen of the MI in the occipital lobe are linked

I suggest dedicating more place in the paper to the neurocognitive aspect and the hypothesis possibly associated with this finding than to the need of improving the epileptogenic biomarkers. Indeed, the HFO and MI analysis is done on the “physiological” channels, so no information is done on the real identification of the EZN in these patients. If you claim this, you should also analyse the epileptogenic channels.

For example, in the abstract and introduction you mention that this atlas could improve the “age-appropriate localization of the epileptogenic zone”, I’m not sure that your results can assure this, it is more an adding information on physiological developmental activities as no data support a better delineation of the EZN in these patients.

From a practical point of view, I find difficult to use these results in the clinical practice for the definition of the epileptogenic zone, but they are really important for the understanding of the visual memory process. As mentioned in the paper, HFO in the occipital lobe are found at a very high frequency rate in the physiological brain and it is difficult to use these markers for the definition of the EZN in this area.

The authors analysed a large cohort of patients (n=114) including very young children (n= 14 < 4 years old) studied by iEEG and operated with an Engel class I in the follow up. The brain area analysed were the “physiological” ones, so the ones not involved in the EZN.

Some additional analysis could strengthen the actual results:

1. It will be of interest to compare the MI in the “physiological” occipital lobe with the MI in the “pathological” ones and possibly with neuropsychological evaluations. Does it exist a “threshold” of MI

that distinguishes the physiological from pathological channels? Do neurophysiological data correlate with neuropsychological examination?

2. If MI is involved in visual memory consolidation, children with visual memory impairment may possibly show modifications in this index. I think more data is needed to support your hypothesis on visual memory.

3. I found it limiting not to define better the criteria of "slow wave sleep". In animals, there is no clear consensus but in humans, it should correspond to the criteria of slow wave sleep N3 according to the AASM (American academy of sleep medicine) and specify if it is the first or second part of the night at least because the proportion of slow waves is not the same. It would also be necessary to specify on which way they determined the stage of sleep, knowing that the amplitude of the slow waves can be different according to the regions.

4. Data are collected during sleep; do you think that dreams rather than the sleep modifications during this early stage of life can play a role? Did you look to spindles activities? Could be MI involved in such a change? Occipito-parietal regions are also thought to be involved in consciousness (Siclari et al., Nat Neurosci 2017), do you think your findings could be somehow linked to such experiences in the young child?

5. From the videos, it seems that the mesial part of the fronto-parietal lobes behaves differently from the lateral ones, did you try to separate these areas in your statistical analysis? In particular, hippocampus is known to have physiological ripples especially during sleep involved in memory processing, so maybe it deserves a different approach that being included in all the temporal lobe.

The authors share with the community all data and analysis code permitting reproducibility of the results. Methods are well described and signal analysis is rigorous, with a comparison of different methods of HFOs detections (open source: ripple lab). The match of neurophysiological findings with MRI DWI improve the quality of the results and represent an important step towards a multidisciplinary approach to physiology. However, a more detailed methodological analysis should be done and it is not in my expertise (in particular MRI tractography analysis and statistical analysis).

Limitations are represented by the subdural iEEG technique that evaluates more the cortical aspects of the brain, without a good sampling i.e. of the mesial temporal regions that are fundamental for memory consolidation; this limits to me the word "global atlas" because some important regions are not sufficiently sampled.

Minor comments

I could not find the ancillary analysis of the rates of HFO >150 Hz and 250 Hz that could indeed be of interest regarding the difference between pathological and physiological HFO even if done on a smaller cohort of patients.

Regarding occipital HFA independently of epileptogenic area, you can cite also Melani 2013 Continuous High Frequency Activity: A peculiar SEEG pattern related to specific brain regions.

What do you mean with the word "enhanced MI?" could you better clarify?

Reviewer #4 (Remarks to the Author):

In this paper, the authors develop a normative atlas for phase-amplitude coupling (via the modulation index, MI) and high frequency oscillations (HFO) in 114 pediatric patients implanted with iEEG. They report several significant relationships between age and MI, as well as age and HFOs, and they combine this with white matter tractography, which provides evidence of connectivity between the associated regions. The large dataset, the testing of multiple frequency band pairs for MI, and the use of four different HFO detectors are strengths. The focus on a pediatric patient population and the combination of computational iEEG analysis with white matter tractography are novel. However, I have several significant methodological concerns, as detailed below.

1) The inclusion of only four (out of 114) patients above 20 years old is concerning, as the three 30-40 year-old-patients are outliers in terms of age and may have an outsized influence on the regression results. Because the study focuses on a pediatric patient population, it seems the authors would be justified in excluding these adult subjects. At the very least, I would suggest that the authors redo all regression results after excluding those four subjects, to verify that the significant results still hold.

2) It is difficult to see evidence of the reported changes in MI and HFOs with age, as shown in Figure 2. I appreciate that all data points are shown, but there are so many data points and the figures are so small that the relationships with age are not evident. Perhaps converting these to violin plots or boxplots (individual plots for each year of age, or each two years of age) would be more convincing, so the reader could better interpret the distribution of values at each age?

Minor comments:

3) The WinPACT toolbox can calculate phase-amplitude coupling using several different methods. Which one was used here?

4) The first two paragraphs on p.18 seem to repeat the same information and could be combined into one paragraph.

5) The data in Figure 5A are difficult to see, as the figures are very dark.

6) Some clarification on the white matter tract results (p.21-22) would be helpful. My interpretation is that a total of 9464 mesh points were tested across the entire brain, and 159 showed a significant

relationship between the square root of age and the MI value. Some of these 159 points were deemed to be connected based on the DWI analysis; it would be helpful if the authors gave the exact numbers of pairs/connections. I have the same question for the 902 mesh points associated with HFO co-diminution.

7) In the first paragraph of the discussion, it would be helpful to know what a typical MI value is in epileptogenic cortex, especially in the occipital lobe, if it is known. I am wondering if the high occipital lobe values in non-epileptogenic cortex are really large enough to be mistaken as pathological.

8) Similarly, in the second section of the discussion (on clinical significance of the HFO results), it would be helpful to know how the ranges of HFO rates reported here compare to the adult normative atlases that have been developed. Do the values for the older children approach those of adults?

REVIEWER COMMENTS

Reviewer #1 (Remarks to the Author):

[Comment 1] The authors recorded intracranial EEG (iEEG) during the presurgical workup in patients with epilepsy. There are several biomarkers in the iEEG to guide epilepsy surgery. Currently, high frequency oscillations (HFOs) are proposed as a new biomarker as well as their relation to slow waves (MI). The authors studied HFO and MI in brain healthy tissue, far from the supposed epileptogenic zone. One motivation of the study is to provide a normative atlas of background activity against which pathological activity can be identified. A second motivation is to present the development of MI with age during childhood, where MI may be a marker of visual memory consolidation. The number of patients is fairly large. The methods use standard toolboxes for iEEG time series analysis and image processing. This study extends a previous similar publication from the MNI group (Frauscher et al., Ann Neurol 2018) to patients during their childhood. The data analysis seems solid. The interpretation of MI as indicator for visual memory consolidation is somewhat speculative: The visual memory consolidation was not tested in these patients but only inferred from the literature.

[Response 1] We sincerely thank Reviewer #1 for the valuable and constructive feedback. We concur with Reviewer #1's perspective, highlighting the necessity to conduct an experiment incorporating a pre-sleep visual task and a subsequent behavioral test upon awakening.

Several previous studies, including Landsness et al. (Sleep, 2009) with 12 healthy participants using scalp EEG, Yotsumoto et al. (Curr Biol, 2009) involving 21 healthy participants using fMRI, and Zhang et al. (Nat Commun, 2018) studying 12 patients through invasive EEG sampling predominantly from the temporal lobe regions, have employed suitable experimental designs to determine the significance of EEG biomarkers in visual memory consolidation. Landsness et al. (2009) illustrated that the performance of a visuo-motor task post-awakening was adversely affected by the acoustic stimuli-induced suppression of sleep slow waves. Yotsumoto et al. (2009) observed a correlation between the enhanced performance in a task involving visual perceptual learning and the level of trained-region-specific hemodynamic activation in the primary visual cortex. Zhang et al. (2018) found that the gamma activity_{30-90 Hz} in the temporal lobe, when replayed during sleep, was linked with successful recall upon awakening. By employing such a suitable study design, one could potentially confirm a causal link between visual memory consolidation and occipital MI, capable of rating delta-nested HFO during sleep.

In our study, however, none of our study patients were given a pre-sleep visual task followed by a behavioral test upon awakening. Thus, we were only able to assess the correlation between the z-score normalized occipital MI recorded intracranially and the baseline neuropsychological data compiled before iEEG recording. We employed an additional mixed model analysis applied to 48 patients who underwent Peabody Picture Vocabulary Test, and this analysis failed to find a significant association between z-score normalized occipital MI and neuropsychological score. Thereby, normalized MI reflects a value normalized to the mean expected to a given site in a given age group.

In response to [Comment 12], our revised manuscript has moderated the intensity of the assertion that occipital delta-nested HFO assists visual memory consolidation throughout the main text. For example, we have removed the statements regarding visual memory consolidation from the abstract. We have also underscored in the discussion that the causal relationship between

occipital delta-nested HFO and visual memory consolidation remains a hypothesis for future testing.

In the Results section, we have provided the following paragraph: “***Relationship between MI and neuropsychological score*** A total of 50 patients underwent Peabody Picture Vocabulary Test (PPVT) (Dunn and Dunn, 2007) prior to surgery; 48 out of the 50 patients had iEEG sampling from the occipital lobe, so they were included in the following statistical analysis. Mixed model analysis failed to show a significant association between z-score normalized $MI_{\geq 80}$ Hz & 0.5-1 Hz and PPVT score (mixed model effect: 2.4×10^{-9} ; uncorrected p-value: 1.00; t-value: 4.4×10^{-5} ; DF: 453).”

In the Discussion section, we have provided the following paragraph: “In the current study, we found no significant association between the z-score normalized $MI_{\geq 80}$ Hz & 0.5-1 Hz in the occipital lobe and the age-normalized PPVT score, an instrument measuring visual object recognition. Given that we failed to find an association between MI and PPVT, one must be cautious in extrapolating a causal relationship between delta-nested HFO and visual memory consolidation. Previous research has employed more targeted methodologies to discern the impact of EEG measures on visual memory consolidation (Landsness et al., 2009; Yotsumoto et al., 2009; Zhang et al., 2018). These studies incorporated a pre-sleep visual task, followed by a behavioral test upon awakening—a process that may more effectively reveal the causal relationship between cortical signals and visual memory consolidation. A study of 12 patients through invasive EEG sampling predominantly from the temporal lobe regions reported that gamma activity₃₀₋₉₀ Hz in the temporal lobe, when replayed during sleep, was associated with successful recall upon awakening (Zhang et al., 2018). Conversely, our study did not provide any of the participating patients with a pre-sleep visual task or a post-sleep behavioral test. A study with a proper visual perceptual task (Yotsumoto et al., 2009) and behavioral assessment before and after sleep is necessary to better determine the causal relationship between occipital delta-nested HFO and visual memory consolidation.”

In the Methods section, we have provided the following paragraph: “***Relationship between MI and neuropsychological score*** To explore the relationship between occipital MI and visual object recognition skills, we employed mixed model analysis on patients who had undergone the PPVT (Dunn and Dunn, 2007) before surgery and had at least one nonepileptic electrode site within the occipital lobe. The dependent variable was the baseline PPVT score, and the fixed effect predictor was z-score normalized $MI_{\geq 80}$ Hz & 0.5-1 Hz. The random effect factors included intercept and patient.”

[Comment 2] The conclusions with respect to epilepsy surgery should be more cautious.

[Response 2] We employed additional mixed model analyses to address this comment. The analysis revealed that electrode contacts located within the seizure onset zone (SOZ) exhibited significantly higher z-score normalized modulation index (MI) and rates of high-frequency oscillation (HFO) compared to those within nonepileptic sites in each lobe. These findings provide support for the hypothesis that statistical deviations from normative means at given sites and ages could aid in localizing the epileptogenic zone during presurgical evaluation of epilepsy. Considering the recommendation made in Chen et al., Neurology 2021, we have cautiously formulated our conclusion as follows: "We have publicly shared our intracranial EEG data to enable investigators to validate MI and HFO-centric presurgical evaluations to identify the epileptogenic zone." Please note that the abstract should not exceed a word count of 150 words.

We have found that some studies published in this journal made such a cautious conclusion in the abstract (e.g., Nat Commun 2022;13:994).

In the Results section we have provided the following statements:

“Statistical deviation of MI and HFO in the seizure onset zone (SOZ)

In **Supplementary Movies 6 and 7**, we have illustrated the normative mean plus two standard deviations of MI biomarkers, as indicated by the regression model, to help readers understand the typical ranges of these markers for given age groups. Furthermore, mixed model analysis employed to each of the four brain lobes demonstrated that z-score normalized MI, reflecting the statistical deviation of MI from the normative mean at given sites and ages, was significantly higher in the SOZ compared to nonepileptic sites. The aforementioned statistical statement was applicable to each of the following MI biomarkers: $MI_{\geq 80 \text{ Hz and } 0.5-1 \text{ Hz}}$, $MI_{\geq 80 \text{ Hz and } 3-4 \text{ Hz}}$, $MI_{\geq 150 \text{ Hz and } 0.5-1 \text{ Hz}}$, and $MI_{\geq 150 \text{ Hz and } 3-4 \text{ Hz}}$ (**Table 4**). The exception was that $MI_{\geq 150 \text{ Hz and } 0.5-1 \text{ Hz}}$ did not differ between the SOZ and nonepileptic sites within the occipital lobe.

Supplementary Movie 8 indicates the mean plus two standard deviations of $HFO_{HIL \geq 80 \text{ Hz}}$ rate for given age groups. Mixed model analysis indicated that z-score normalized $HFO_{HIL \geq 80 \text{ Hz}}$ was higher in the SOZ compared to nonepileptic sites (**Table 4**). Due to the infrequent occurrence of events, we were unable to compute the statistical deviation of $HFO_{HIL \geq 150 \text{ Hz}}$ in a meaningful manner.”

Biomarker	Lobe	Mixed model estimate	SE	t-value	DF	p-value	Lower 95% CI	Upper 95% CI
$MI_{\geq 80 \text{ Hz and } 0.5-1 \text{ Hz}}$	Frontal	0.75	0.070	10.8	3238	< 0.001	0.61	0.89
	Temporal	3.00	0.48	6.25	2811	< 0.001	2.06	3.94
	Parietal	1.27	0.11	11.8	2242	< 0.001	1.06	1.48
	Occipital	0.45	0.14	3.17	958	0.0016	0.17	0.73
$MI_{\geq 80 \text{ Hz and } 3-4 \text{ Hz}}$	Frontal	3.52	0.18	19.7	3238	< 0.001	3.17	3.87
	Temporal	4.10	0.33	12.4	2811	< 0.001	3.45	4.75
	Parietal	5.82	0.30	19.1	2242	< 0.001	5.22	6.42
	Occipital	1.83	0.21	8.73	958	< 0.001	1.42	2.24
$MI_{\geq 150 \text{ Hz and } 0.5-1 \text{ Hz}}$	Frontal	0.51	0.055	9.30	3238	< 0.001	0.40	0.62
	Temporal	2.56	0.81	3.14	2811	0.0017	0.96	4.15
	Parietal	0.62	0.065	9.50	2242	< 0.001	0.49	0.75
	Occipital	0.079	0.12	0.67	958	0.50	-0.15	0.31
$MI_{\geq 150 \text{ Hz and } 3-4 \text{ Hz}}$	Frontal	3.70	0.30	12.5	3238	< 0.001	3.12	4.28
	Temporal	3.77	0.73	5.16	2811	< 0.001	2.34	5.20
	Parietal	5.41	0.43	12.5	2242	< 0.001	4.56	6.26
	Occipital	0.94	0.18	5.11	958	< 0.001	0.58	1.31
$HFO_{HIL \geq 80 \text{ Hz}}$	Frontal	2.45	0.079	30.9	3238	< 0.001	2.30	2.61
	Temporal	1.85	0.080	23.3	2811	< 0.001	1.70	2.01

	Parietal	2.95	0.10	28.7	2242	< 0.001	2.75	3.16
	Occipital	1.59	0.14	11.4	958	< 0.001	1.32	1.86

Table 4. Comparison of intracranial EEG biomarker values between the seizure onset and nonepileptic sites. The results of mixed model analyses are provided with uncorrected p-values. All z-score normalized biomarker values mentioned in this table (except $MI_{\geq 150 \text{ Hz}}$ and $0.5\text{-}1 \text{ Hz}$) were significantly higher in the seizure onset than in the nonepileptic sites. CI: 95% confidence interval. DF: degree of freedom. SE: standard error. $MI_{\geq f \text{ Hz and } s \text{ Hz}}$ denotes the strength of coupling between the amplitude of $HFO_{\geq f \text{ Hz}}$ and the phase of slow-wave_s Hz. $HFO_{HIL \geq 80 \text{ Hz}}$ denotes the rate of high-frequency oscillation defined by the Hilbert method. We could not compute z-score normalized $HFO_{HIL \geq 150 \text{ Hz}}$ because the regression model failed to fit the data of $HFO_{HIL \geq 150 \text{ Hz}}$ rates in a meaningful manner, due to the infrequent occurrence of events.

In **Supplementary Movies 6, 7, and 8**, we have provided information useful to understand the normative ranges of MI and $HFO_{HIL \geq 80 \text{ Hz}}$. Below, we have provided the snapshots of **Supplementary Movies 6, 7, and 8**.

Supplementary Movie 6. The normative ranges of cortical $MI \geq 80$ Hz and 0.5-1 Hz and $MI \geq 80$ Hz and 3-4 Hz.

On the left side, the video presents the mean of normative MI at given mesh points, as estimated by the univariate regression model incorporating $\sqrt{\text{age}}$. On the right side, it presents the mean plus two standard deviations, likewise estimated by the univariate regression model. For children who are 'n' years old, we computed the standard deviation across children between 'n' and 'n+3.9' years old. The brain images in this movie were created using FreeSurfer (<https://surfer.nmr.mgh.harvard.edu/fswiki/CorticalParcellation>).

Timeline

00:00-00:25 $MI > 80$ Hz and 0.5-1 Hz.

00:25-00:50 $MI > 80$ Hz and 3-4 Hz.

Supplementary Movie 7. The normative ranges of cortical $MI \geq 150$ Hz and 0.5-1 Hz and $MI \geq 150$ Hz and 3-4 Hz. On the left side, the video presents the mean of normative MI at given mesh points, as estimated by the univariate regression model incorporating $\sqrt{\text{age}}$. On the right side, it presents the mean plus two standard deviations, likewise estimated by the univariate regression model. For children who are 'n' years old, we computed the standard deviation across children between 'n' and 'n+3.9' years old. The brain images in this movie were created using FreeSurfer (<https://surfer.nmr.mgh.harvard.edu/fswiki/CorticalParcellation>).

Timeline

00:00-00:25 $MI > 150$ Hz and 0.5-1 Hz.

00:25-00:50 $MI > 150$ Hz and 3-4 Hz.

Supplementary Movie 8. The normative range of cortical $HFO_{HIL} \geq 80$ Hz. On the left side, the video presents the mean of normative $HFO_{HIL} > 80$ Hz at given mesh points, as estimated by the univariate regression model incorporating $\sqrt{\text{age}}$. On the right side, it presents the mean plus two standard deviations, likewise estimated by the univariate regression model. For children who are 'n' years old, we computed the standard deviation across children between 'n' and 'n+3.9' years old. The brain images in this movie were created using FreeSurfer (<https://surfer.nmr.mgh.harvard.edu/fswiki/CorticalParcellation>).

In the Discussion section, we have provided the following statements: “In the present study, we have provided atlases presenting the expected mean plus two standard deviations of MI biomarkers to illustrate the typical ranges for given age groups (**Supplementary Movies 6 and 7**). We believe our atlas has the potential to serve as a valuable reference in presurgical evaluations, as z-score normalized MI values were significantly higher in the SOZ compared to nonepileptic sites (**Table 4**). Compared to $MI > 80$ Hz & 0.5-1 Hz, $MI > 80$ Hz & 3-4 Hz showed much larger effect sizes of difference between the SOZ and nonepileptic sites, as suggested by the mixed model estimate (**Table 4**). This observation can be attributed to the notion that interictal epileptiform discharges are generally associated with a transient increase in HFO stereotypically coupled with a delta wave at 3-4 Hz⁶, whereas physiological HFO cycles between augmentation and attenuation during slow-wave sleep at < 1 Hz¹⁸⁻²⁰. A prospective study is warranted to investigate whether a more inclusive resection of cortical sites, whose $MI > 80$ Hz & 3-4 Hz values deviate from the age-specific normal range, would predict better postoperative seizure control in young children.”

“**Supplementary Movie 8** presents the mean plus two standard deviations of the $HFO_{HIL} > 80$ Hz rate, as suggested by the regression model. This atlas may be useful in helping investigators understand the typical range of this biomarker value at given sites and given ages. Our mixed

model analysis demonstrated that, within each brain lobe, z-score normalized $HFO_{HIL}>80$ Hz was higher in the SOZ compared to nonepileptic sites (**Table 4**).

In the Methods section, we have provided the following paragraph:
“MI and HFO in the seizure onset zone (SOZ) We employed mixed model analyses in given brain lobes to determine whether the SOZ electrode sites exhibited significantly higher MI and HFO rates compared to nonepileptic sites. Here, the dependent variable was z-score normalized $MI>80$ Hz and 0.5-1 Hz, $MI>80$ Hz and 3-4 Hz, $MI>150$ Hz and 0.5-1 Hz, $MI>150$ Hz and 3-4 Hz, $HFO_{HIL}>80$ Hz, and $HFO_{HIL}>150$ Hz at a given electrode site. We computed a z-score normalized value of a given iEEG biomarker using the mean and standard deviation across 30 nonepileptic electrode sites closest to a given electrode contact (Kuroda et al., 2021); thereby, for children who are 'n' years old, we computed the mean and standard deviation across children between 'n' and 'n+3.9' years old. The fixed effect predictor included the SOZ label (SOZ = 1; nonepileptic = 0). The random effect factors included the intercept and patient. We deemed an FDR-corrected p-value of 0.05 (for six comparisons: six iEEG measures) as the significance threshold.

We have illustrated the mean plus two standard deviations of the aforementioned iEEG biomarkers, as indicated by the regression model, to help readers understand the typical ranges of these markers for given age groups (**Supplementary Movies 6-8**).

Our findings provide support for the hypothesis that statistical deviations from normative means at given sites and in given ages could aid in localizing the epileptogenic zone during presurgical evaluation of epilepsy.

[Comment 3] In particular, the manuscript must be improved in several ways. 1. The iEEG figures must be improved. All the values given in the legend should be presented in a table and not in the figure panel.

[Response 3] We have presented the results of regression and mixed model analyses in the form of tables as well. As a result, we have provided 43 new tables as part of the revised manuscript (i.e., **Tables 2, 3 and 4; Supplementary Tables 2-41**).

[Comment 4] The authors use the same y-axis in all plots for better comparison across brain areas. This puts many points close to $y=0$ so that they cannot be discerned. I suggest a \log_{10} y-scale combined with a \log_{10} x-scale. Then the fit of a power law will appear as a straight line with the slope being the exponent. Given the three linear regressions used in this study (age, square root of age, and \log_{10} age), this will be more illustrative. At the same time, it will allow a visual judgement of the regression, which is not possible now for many subplots.

[Response 4] To improve the visibility of data points, we have converted dot plots to violin plots, as suggested by the author guidelines. Since HFO rates were zero/min in many electrode sites, we are unable to employ a \log_{10} y-scale. The new figures are provided below.

Figure 2. The developmental changes of cortical MI and HFO at given lobes. a $MI_{\geq 80 \text{ Hz} \ \& \ 0.5-1 \text{ Hz}}$ denotes the strength of coupling between the amplitude of $HFO_{\geq 80 \text{ Hz}}$ and the phase of slow-wave $0.5-1 \text{ Hz}$, as rated by modulation index. **b** Occurrence rate (/min) of $HFO_{HIL \geq 80 \text{ Hz}}$ as defined by the Hilbert method. In each violin plot, a regression line is provided based on a model incorporating the square root of age ($\sqrt{\text{age}}$) as an independent variable. The white circle within each violin plot represents the median.

Supplementary Figure 3. The developmental changes of cortical MI and HFO at given lobes.
a $MI_{\ge 80 \text{ Hz} \& 3-4 \text{ Hz}}$: the strength of coupling between the amplitude of $HFO_{280 \text{ Hz}}$ and the phase of slow-wave $_{3-4 \text{ Hz}}$, as rated by modulation index. **b** $HFO_{STE \ge 80 \text{ Hz}}$ occurrence rate (/min). **c** $HFO_{SLL \ge 80 \text{ Hz}}$ occurrence rate. **d** $HFO_{MNI \ge 80 \text{ Hz}}$ occurrence rate. In each violin plot, a regression line is provided based on a model incorporating the square root of age ($\sqrt{\text{age}}$) as an independent variable. The white circle within each violin plot represents the median. MI: modulation index. HFO_{STE} : high-frequency oscillation (HFO) defined by Staba et al., 2002. HFO_{SLL} : HFO defined by Gardner et al., 2007. HFO_{HIL} : HFO defined by Crépon et al., 2010. HFO_{MNI} : HFO defined by Zelmann et al., 2010.

Supplementary references.

- Staba, R. J., Wilson, C. L., Bragin, A., Fried, I. & Engel, J. Jr. Quantitative analysis of high-frequency oscillations (80-500 Hz) recorded in human epileptic hippocampus and entorhinal cortex. *J Neurophysiol.* **88**, 1743-1752 (2002).
- Gardner, A. B., Worrell, G. A., Marsh, E., Dlugos, D. & Litt, B. Human and automated detection of high-frequency oscillations in clinical intracranial EEG recordings. *Clin Neurophysiol.* **118**, 1134-1143 (2007).
- Crépon, B. et al. Mapping interictal oscillations greater than 200 Hz recorded with intracranial macroelectrodes in human epilepsy. *Brain.* **133**, 33-45 (2010).
- Zelmann, R. et al. Automatic detector of high frequency oscillations for human recordings with macroelectrodes. *Annu Int Conf IEEE Eng Med Biol Soc.* **2010**, 2329-2333 (2010).

[Comment 5] 3. The three patients >20years appear as outliers. Please redo all analyses without these three patients to verify that you results are valid for your core set of patients, which is aged ≤20years.

[Response 5] We performed ancillary mixed model analyses, excluding three patients of 21 years old and above. We have verified that the results of these analyses do not change our conclusions.

For example, as shown in **Table 2 and Supplementary Table 2** below, the developmental growth (slope/'Iyear) of MI was prominent in the occipital lobe and modest, if any, in the remaining brain lobes.

Lobe	Slope (/'Iyear)	Uncorrected p-value	t-value	DF	Lower 95% CI	Upper 95% CI
Frontal	7.0×10^{-3}	$*3.0 \times 10^{-6}$	4.7	2978	4.1×10^{-3}	9.9×10^{-3}
Temporal	7.2×10^{-3}	$*6.5 \times 10^{-8}$	5.4	2393	4.6×10^{-3}	9.7×10^{-3}
Parietal	0.011	$*2.4 \times 10^{-9}$	6.0	1999	7.4×10^{-3}	0.015
Occipital	0.047	$*3.1 \times 10^{-12}$	7.1	873	0.034	0.060

Table 2. The results of regression analysis to assess the effect of 'Iage on $MI_{\geq 80 \text{ Hz}} \& 0.5-1 \text{ Hz}$. $MI_{\geq 80 \text{ Hz}}$ and $0.5-1 \text{ Hz}$ denotes the strength of coupling between the amplitude of high-frequency oscillation $>80 \text{ Hz}$ and the phase of slow-wave $0.5-1 \text{ Hz}$. CI: confidence interval. DF: degree of freedom. *: significant with False Discovery Rate (FDR) correction. **Supplementary Table 2** presents the results of the analysis, excluding three patients of 21 years old and above.

Lobe	Slope (/'Iyear)	Uncorrected p-value	t-value	DF	Lower 95% CI	Upper 95% CI
Frontal	5.8×10^{-3}	$*6.9 \times 10^{-4}$	3.4	2895	2.5×10^{-3}	9.2×10^{-3}
Temporal	5.1×10^{-3}	$*3.0 \times 10^{-4}$	3.6	2314	2.4×10^{-3}	7.9×10^{-3}
Parietal	8.2×10^{-3}	$*3.0 \times 10^{-5}$	4.2	1949	4.4×10^{-3}	0.012
Occipital	0.049	$*4.1 \times 10^{-11}$	6.7	831	0.034	0.063

Supplementary Table 2. The results of ancillary regression analysis to assess the effect of 'Iage on $MI_{\geq 80 \text{ Hz}} \& 0.5-1 \text{ Hz}$ in a given brain lobe. Here, we present the results of ancillary analyses, excluding three patients of 21 years old and above. $MI_{\geq 80 \text{ Hz}}$ and $0.5-1 \text{ Hz}$ denotes the strength of coupling between the amplitude of high-frequency oscillation $>80 \text{ Hz}$ and the phase of slow-wave $0.5-1 \text{ Hz}$. CI: confidence interval. DF: degree of freedom. *: significant with False Discovery Rate (FDR) correction.

We also provided the results of mixed model analyses, with all 114 patients included (**Supplementary Tables 4-7**) and with three patients of >21 years old excluded (**Supplementary Tables 8-11**). Both models commonly indicated that the effect of 'Iage on MI was significant in the occipital lobe but not in the remaining brain lobes.

As shown in **Table 3 and Supplementary Table 3** below, the developmental diminution (slope/ $\sqrt{\text{year}}$) of HFO_{HIL} >80 Hz rate was prominent in the frontal, temporal, and parietal lobes, regardless of excluding three patients of 21 years old and above.

Lobe	Slope (/ $\sqrt{\text{year}}$)	Uncorrected p-value	t-value	DF	Lower 95% CI	Upper 95% CI
Frontal	-0.41	*4.3 x 10 ⁻⁵⁴	-15.8	2978	-0.46	-0.36
Temporal	-0.30	*7.4 x 10 ⁻²⁸	-11.0	2393	-0.36	-0.25
Parietal	-0.33	*3.7 x 10 ⁻²¹	-9.5	1999	-0.40	-0.26
Occipital	0.081	0.31	1.0	873	-0.075	0.24

Table 3. The results of regression analysis to assess the effect of 'Age on HFO_{HIL} \geq 80 Hz occurrence rate. HFO_{HIL} >80 Hz: high-frequency oscillation >80 Hz defined by the Hilbert method. CI: confidence interval. DF: degree of freedom. *: significant with False Discovery Rate (FDR) correction. **Supplementary Table 3** presents the results of the analysis, excluding three patients of 21 years old and above.

Lobe	Slope (/ $\sqrt{\text{year}}$)	Uncorrected p-value	t-value	DF	Lower 95% CI	Upper 95% CI
Frontal	-0.45	*2.7 x 10 ⁻⁴⁹	-15.0	2895	-0.51	-0.39
Temporal	-0.38	*1.6 x 10 ⁻³⁴	-12.5	2314	-0.44	-0.32
Parietal	-0.39	*4.9 x 10 ⁻²⁵	-10.5	1949	-0.46	-0.32
Occipital	0.13	0.15	1.4	831	-0.046	0.30

Supplementary Table 3. The results of ancillary regression analysis to assess the effect of 'Age on HFO_{HIL} \geq 80 Hz occurrence rate in a given brain lobe. Here, we present the results of ancillary analyses, excluding three patients of 21 years old and above. HFO_{HIL} >80 Hz: high-frequency oscillation >80 Hz defined by the Hilbert method. CI: confidence interval. DF: degree of freedom. *: significant with False Discovery Rate (FDR) correction.

Furthermore, we provided the results of mixed model analyses, with all 114 patients included (**Supplementary Tables 34-37**) and with three patients of >21 years old excluded (**Supplementary Tables 38-41**). Both models commonly indicated that the developmental diminution of HFO_{HIL} >80 Hz rate was significant in the frontal, temporal, and parietal lobes but not in the occipital lobe.

[Comment 6] The authors record iEEG with 1000 Hz. This precludes analysis of spectral content >250Hz. All their text on HFO >250Hz (Fast Ripples) is unfounded speculation not based on their data. This text should simply be removed.

[Response 6] We have decided not to provide the analysis of HFO >250 Hz, as suggested.

[Comment 7] Please read the reference (Zweiphenning et al., 2022a) closely and summarize it correctly. The current summary is wrong.

[Response 7] Thank you very much. We have provided the revised statement as follows: “A randomized clinical trial reported that the efficacy of intraoperatively measured HFO-guided resection in controlling seizures was non-inferior to that of conventional resection guided by interictal spike discharges in patients with extra-temporal lobe epilepsy but not in those with temporal lobe epilepsy. (Zweiphenning et al., 2022a).”

[Comment 8] Please remove the older reviews on HFO and replace them by (Chen Z, Maturana MI, Burkitt AN, Cook MJ, Grayden DB. High-frequency Oscillations in Epilepsy: What Have We Learned and What Needs to be Addressed. *Neurology*. 2021. <https://doi.org/10.1212/wnl.00000000000011465>). I recommend reading that review closely. While the study analyses four types of HFO for which detectors are available in the toolbox, their value for epilepsy surgery is still debated.

[Response 8] Thank you very much. We have cited Chen et al., *Neurology* 2021 twice, while removing all of the older review articles. We have formulated our conclusion cautiously considering the conclusions and recommendations made in Chen et al., *Neurology* 2021.

[Comment 9] While the conclusions regarding epilepsy appear in the abstract, they should be made explicit at the end of the manuscript. Of course, they must be cautious and limited to the data provided in this study. Explicit conclusions will help the reader to understand the noteworthiness of the study.

[Response 9] We have provided the following conclusion statement in the end of the discussion: “In summary, our study of 114 patients achieving ILAE class-1 outcomes demonstrated that z-score normalized MI and HFO_{≥HIL80} Hz rate were significantly higher within the SOZ compared to nonepileptic sites in each lobe. A prospective study is warranted to investigate whether a more inclusive resection of cortical sites, whose iEEG biomarker values deviate from the age-specific normal range, would predict better postoperative seizure control in young children.”

Please note that we have revised our manuscript entirely so that it complies with the *Nature Communications* formatting instructions. The specific changes include: [1] reformatting the list of authors; [2] shortening the abstract to less than 150 words; [3] removal of subheadings from the introduction; [4] starting the final paragraph of the introduction with the phrase “In this study, our normative atlases reveal ...”; [5] removal of numbering from the subheading; [6] removal of subheadings from the discussion; [7] revision of the data availability statement; [8] revision of the code availability statement; [9] reformatting of reference citations; [10] reduction of the number of references to no more than 70; [11] reformatting of the author contributions; [12] reformatting of **Figures 1-5**; [13] reformatting of **Tables 1-4**; [14] reformatting of the supplementary information (**Supplementary Tables 1 – 41**, **Supplementary Figures 1 – 4**, and Legends for **Supplementary Movies 1 - 8**); [15] conversion of the supplementary information to a PDF document; [16] reformatting of the main text; and [17] reduction of the number of words in the main text (not including figure legends or Methods) to 6000.

[Comment 10] Please label the video file names for easier access.

[Response 10] Each movie file shows a given video file name in the beginning (e.g., **Supplementary Movie 1**).

Reviewer #2 Maeike Zijlmans (Remarks to the Author):

[Comment 11] This paper describes the presumed physiological activity (HFOs and MI) in 114 patients with focal epilepsy using invasive EEG electrodes outside the diseased areas. It creates a topographical and age-related 4-dimensional atlas. I think this is a very nice and relevant attempt, yielding a beautiful data representation useful for researchers in neuroscience from many directions. The 4-dimensionality is novel. The work supports the conclusions and claims. I did not notice evident flaws. The methodology is generally sound and enough detail is provided. I have two general and several specific comments. General 1) The writing is generally good, but I believe it can be much more structured (adhering strictly to intro-methods-results sections and use the same structures within). I will mention some examples in my comments below.

[Response 11] We appreciate Dr. Zijlmans' thoughtful comments and recommendations. Based on the feedback received, we have taken steps to enhance the structure of the manuscript. Specifically, we have removed the methodological statements from the introduction and relocated most of the statements about the results from the methods section. Following the *Nature Communications* formatting instructions, we needed to restructure the main text to adhere to their guidelines. The revised manuscript now follows the sequence of **[Introduction]**, **[Results]**, **[Discussion]**, and **[Methods]**, as specifically requested. Furthermore, the revised introduction needs to provide a brief summary of the major results and conclusions of the current work, as suggested in the author guidelines. Taken together, we have revised our manuscript entirely so that it complies with the *Nature Communications* formatting instructions. The specific changes include: [1] reformatting the list of authors; [2] shortening the abstract to less than 150 words; [3] removal of subheadings from the introduction; [4] starting the final paragraph of the introduction with the phrase “In this study, our normative atlases reveal ...”; [5] removal of numbering from the subheading; [6] removal of subheadings from the discussion; [7] revision of the data availability statement; [8] revision of the code availability statement; [9] reformatting of reference citations; [10] reduction of the number of references to no more than 70; [11] reformatting of the author contributions; [12] reformatting of **Figures 1-5**; [13] reformatting of **Tables 1-4**; [14] reformatting of the supplementary information (**Supplementary Tables 1 – 41**, **Supplementary Figures 1 – 4**, and Legends for **Supplementary Movies 1 - 8**); [15] conversion of the supplementary information to a PDF document; [16] reformatting of the main text; and [17] reduction of the number of words in the main text (not including figure legends or Methods) to 6000.

[Comment 12] 2) Sentences can be shorter. E.g., words like ‘only’ or ‘in this study’ are often redundant and words like ‘as well as’ can be shortened to ‘and’

[Response 12] In the revised manuscript, we have reduced the word count by eliminating redundant words.

[Comment 13] Specific 1) Abstract/manuscript: iEEG included only subdural (electrocorticography) recordings. This should be specified, already in the abstract.

[Response 13] In the abstract, we have specified that iEEG was sampled using subdural electrodes as follows: “We generated normative brain atlases, using subdural EEG signals from 8,251 nonepileptic electrode sites in 114 patients (ages 1.0-41.5 years) who achieved seizure control following resective epilepsy surgery.” Please note that the guidelines of *Nature Communications* limit the abstract to a maximum of 150 words.

[Comment 14] 2) Introduction: ‘independent effects of patient and epilepsy profiles’: how is epilepsy profiles an independent effect? I could not find the definition of ‘epilepsy profiles’ in methods, but it is used several times in abstract, introduction and results.

[Response 14] In the method section, we have specified that the patient demographics considered in the mixed model analysis encompassed age and sex. Additionally, the epilepsy profiles integrated into the mixed model analysis encompassed seizure onset zone (SOZ) location, presence of MRI lesion, sampled hemisphere, and the number of oral antiseizure medications taken immediately before the initiation of intracranial EEG recording.

In the Methods section, specifically, we have provided the following sentence: “We used mixed model analysis (Sonoda et al., 2021; Sakakura et al., 2022) to determine the lobe where a developmental change of a given iEEG biomarker remained significant, after controlling for the independent effects of patient demographics (age and sex) and epilepsy-related variables (SOZ location, MRI lesion, and number of oral antiseizure medications).”

[Comment 15] 3) Introduction: ‘In addition to modeling cortical ontogenic changes, the present study also visualized 140 white matter pathways supporting the development of neural communications via delta141 nested HFO (as rated by $MI \geq 80$ Hz & 0.5-1 Hz), during slow-wave sleep. To accomplish this, we 142 utilized our novel imaging technique, dynamic tractography, which combines iEEG signals 143 with MRI tractography’ This is a methods-like description while in the introduction I want to hear why this extra step was taken and why the specific methods. [Response 15] In the revised manuscript, we have deleted the detailed methodological statements from the introduction but instead we have provided a brief overview of the tractography technique, as shown below: “We thus visualized white matter pathways supporting the development of neural communications via HFO nested in delta waves during slow-wave sleep. To this end, we incorporated MRI tractography and identified the white matter tracts directly connecting cortical regions with significant developmental co-growth of HFO-delta phase-amplitude coupling.”

We have reserved the detailed explanation of the methodology for the dedicated Methods section of the manuscript.

[Comment 16] 4) Methods: a sample rate of 1000 Hz is low to recognize HFOs >250 Hz

[Response 16] We have decided not to provide the analysis of HFO >250 Hz, as suggested by Reviewers #1 and #2.

[Comment 17] 5) Methods/results: the methods describe that 114 patients will be selected and then the inclusion criteria. I want to know how these 114 patients resulted from how many patients in the beginning. It seems that the period is the strict criterium, so than 114 patients is a result from the selection.

[Response 17] We have included **Supplementary Figure 1**, a flowchart that explains how the 114 patients were selected for the study based on the eligibility criteria.

Supplementary Figure 1. Flowcharts showing the study patients satisfying the eligibility criteria.

Out of the 265 patients with a diagnosis of drug-resistant focal epilepsy, 130 patients failed to satisfy the inclusion criteria. Specifically, 107 patients failed to achieve the ILAE class 1 outcome following surgery, whereas 23 patients failed to show an artifact-free 20-minute slow-wave sleep intracranial EEG epoch at least two hours apart from clinical seizure events. Out of the 135 patients satisfying the inclusion criteria, 21 patients were excluded because nine had a history of previous resective surgery, 10 underwent either hemispherotomy or hemispherectomy, and the remaining two had massive brain malformations, making it difficult to identify the central, lateral, or calcarine sulci. As a result, a total of 114 patients satisfying the eligibility criteria were studied.

[Comment 18] 6) Methods/results: To ensure the accuracy of the developmental iEEG 192 atlases, we only included artifact-free iEEG segments from 8,251 nonepileptic electrode sites 193 (mean: 72.4 per patient; range: 12 to 121;) Here again, the number of used/selected channels is a result. I want to know how many channels were discarded because of artefacts, SOZ, MRI, irritative zones.

[Response 18] We have provided the relevant information in the Result section as follows: “We studied 114 patients (ages 1.0 to 41.5 years) who met the eligibility criteria (**Table 1; Supplementary Figure 1**). A total of 8,251 artifact-free nonepileptic electrode sites (mean: 72.4 per patient; range: 12 to 121; **Figure 1**) were available for generating normative atlases. In addition, 1,045 electrode sites within the SOZ (mean: 8.8 per patient; frontal: 260; temporal: 418; parietal: 243; occipital: 85) were available to assess the utility of HFO and MI in distinguishing SOZ from nonepileptic electrode sites. **Supplementary Figure 2** explains how the 8,251 artifact-free nonepileptic electrode sites satisfied the eligibility criteria.”

Supplementary Figure 2 presents flowchart that explains how the 8,251 artifact-free nonepileptic electrode sites were selected for the study based on the eligibility criteria.

Supplementary Figure 2. Flowcharts showing the number of subdural electrodes satisfying the eligibility criteria of artifact-free nonepileptic electrode sites.

The flowchart illustrates the process of identifying the number of subdural electrodes that satisfy the eligibility criteria of being artifact-free nonepileptic electrode sites. Initially, out of the 12708 subdural electrode sites, 1065 sites were affected by artifacts. From the remaining 11643 sites, 1006 were classified as artifact-free seizure onset zone (SOZ) sites. Of the 10637 remaining artifact-free non-SOZ sites, 2,386 were affected by interictal spike discharges or an MRI lesion. Thus, a total of 8,251 sites were considered to be artifact-free nonepileptic electrode sites.

[Comment 19] 7) survived an FDR correction (for 18 comparisons: six iEEG measures \times three types of age measures): I wonder if an FDR is really needed here because both the iEEG measures and the types of age measures are dependant variables. It is good to reduce the change of finding false positive relations but high correction for false discoveries may yield false negative results. This should be discussed in the discussion (Wagenmakers E et al Nature 2022 one statistical analysis must not rule them all).

[Response 19] In the Discussion section, we have provided the following statements: “We determined the age effect on iEEG measures by employing a FDR correction taking into account that we repeated the mixed model analysis 18 times for six iEEG measures and three types of age. This analytic approach reduced the risk of Type I error while increasing the risk of Type II error (Wagenmakers et al., Nature 2022). Thus, the reported absence of statistical difference should be interpreted as failure to find a difference.”

Reviewer #3 (Remarks to the Author):

[Comment 20] I read with interest the paper “Developmental atlas of phase-amplitude coupling between physiologic high-frequency oscillations and slow waves” by Prof Asano and colleagues. In my opinion, the most important result of this paper is the cognitive aspect of the developing brain and the link with MI (an index that evaluates the phase amplitude coupling of HFO and delta waves). In particular, they found that MI is strengthened during development in the occipital lobe and that the cortical regions that show this co-growth are connected by white matter fibers, namely the vertical occipital fasciculi and the posterior callosal fibers. These results could be explained by the visual memory consolidation in early childhood. Moreover, this could be of help in the analysis of the epileptogenic biomarkers sensing to delineate the epileptogenic zone network – EZN - (HFOs and MI), by a better understanding of physiological activities during development. However, I have two main criticisms to this work: 1. the absence of analysis on the pathological channels and the claim of a better definition of the EZN.

From a practical point of view, I find difficult to use these results in the clinical practice for the definition of the epileptogenic zone, but they are really important for the understanding of the visual memory process. As mentioned in the paper, HFO in the occipital lobe are found at a very high frequency rate in the physiological brain and it is difficult to use these markers for the definition of the EZN in this area. For example, in the abstract and introduction you mention that this atlas could improve the “age-appropriate localization of the epileptogenic zone”, I’m not sure that your results can assure this, it is more an adding information on physiological developmental activities as no data support a better delineation of the EZN in these patients. If you claim this, you should also analyse the epileptogenic channels.

Indeed, the HFO and MI analysis is done on the “physiological” channels, so no information is done on the real identification of the EZN in these patients. The authors analysed a large cohort of patients (n=114) including very young children (n= 14 < 4 years old) studied by iEEG and operated with an Engel class I in the follow up. The brain area analysed were the “physiological” ones, so the ones not involved in the EZN. Indeed, the HFO and MI analysis is done on the “physiological” channels, so no information is done on the real identification of the EZN in these patients. Some additional analysis could strengthen the actual results.

[Response 20] We sincerely appreciate the valuable and constructive feedback Reviewer #3 has provided. Please note that we have revised our manuscript entirely so that it complies with the *Nature Communications* formatting instructions. The specific changes include: [1] reformatting the list of authors; [2] shortening the abstract to less than 150 words; [3] removal of subheadings from the introduction; [4] starting the final paragraph of the introduction with the phrase “In this study, our normative atlases reveal ...”; [5] removal of numbering from the subheading; [6] removal of subheadings from the discussion; [7] revision of the data availability statement; [8] revision of the code availability statement; [9] reformatting of reference citations; [10] reduction of the number of references to no more than 70; [11] reformatting of the author contributions; [12] reformatting of **Figures 1-5**; [13] reformatting of **Tables 1-4**; [14] reformatting of the supplementary information (**Supplementary Tables 1 – 41, Supplementary Figures 1 – 4, and Legends for Supplementary Movies 1 - 8**); [15] conversion of the supplementary information to a PDF document; [16] reformatting of the main text; and [17] reduction of the number of words in the main text (not including figure legends or Methods) to 6000.

We have performed additional mixed model analyses to determine how much z-score normalized MI and $HFO_{HIL \geq 80 \text{ Hz}}$ rate were higher in the seizure onset zone (SOZ) as compared to nonepileptic electrode sites. The analysis demonstrated that, in all lobes (including the occipital lobe), z-score normalized MI and $HFO_{HIL \geq 80 \text{ Hz}}$ rate, reflecting the statistical deviation from the mean, were higher within the SOZ compared to nonepileptic sites. This observation supports the potential utility of MI and HFO-based localization of the epileptogenic zone network (Balatskaya et al., Clin Neurophysiol 2020; Bregianni et al., Epilepsy Res 2022), since all study patients achieved ILAE class 1 outcome by definition.

We have provided the results of the mixed model analyses in **Table 4**, as shown below. These mixed model estimates indicate the effect size (akin to Cohen's D) of the given biomarkers in differentiating SOZ from nonepileptic electrode sites within each brain lobe.

Biomarker	Lobe	Mixed model estimate	SE	t-value	DF	p-value	Lower 95% CI	Upper 95% CI
$MI_{\geq 80 \text{ Hz and } 0.5-1 \text{ Hz}}$	Frontal	0.75	0.070	10.8	3238	< 0.001	0.61	0.89
	Temporal	3.00	0.48	6.25	2811	< 0.001	2.06	3.94
	Parietal	1.27	0.11	11.8	2242	< 0.001	1.06	1.48
	Occipital	0.45	0.14	3.17	958	0.0016	0.17	0.73
$MI_{\geq 80 \text{ Hz and } 3-4 \text{ Hz}}$	Frontal	3.52	0.18	19.7	3238	< 0.001	3.17	3.87
	Temporal	4.10	0.33	12.4	2811	< 0.001	3.45	4.75
	Parietal	5.82	0.30	19.1	2242	< 0.001	5.22	6.42
	Occipital	1.83	0.21	8.73	958	< 0.001	1.42	2.24
$MI_{\geq 150 \text{ Hz and } 0.5-1 \text{ Hz}}$	Frontal	0.51	0.055	9.30	3238	< 0.001	0.40	0.62
	Temporal	2.56	0.81	3.14	2811	0.0017	0.96	4.15
	Parietal	0.62	0.065	9.50	2242	< 0.001	0.49	0.75
	Occipital	0.079	0.12	0.67	958	0.50	-0.15	0.31
$MI_{\geq 150 \text{ Hz and } 3-4 \text{ Hz}}$	Frontal	3.70	0.30	12.5	3238	< 0.001	3.12	4.28
	Temporal	3.77	0.73	5.16	2811	< 0.001	2.34	5.20
	Parietal	5.41	0.43	12.5	2242	< 0.001	4.56	6.26
	Occipital	0.94	0.18	5.11	958	< 0.001	0.58	1.31
$HFO_{HIL_{\geq 80 \text{ Hz}}}$	Frontal	2.45	0.079	30.9	3238	< 0.001	2.30	2.61
	Temporal	1.85	0.080	23.3	2811	< 0.001	1.70	2.01
	Parietal	2.95	0.10	28.7	2242	< 0.001	2.75	3.16
	Occipital	1.59	0.14	11.4	958	< 0.001	1.32	1.86

Table 4. Comparison of intracranial EEG biomarker values between the seizure onset and nonepileptic sites. The results of mixed model analyses are provided with uncorrected p-value. All z-score normalized biomarker values mentioned in this table (except $MI_{\geq 150 \text{ Hz and } 0.5-1 \text{ Hz}}$) were significantly higher in the seizure onset zone than in the nonepileptic sites. CI: 95% confidence interval. DF: degree of freedom. SE: standard error. $MI_{\geq f \text{ Hz and } s \text{ Hz}}$ denotes the strength of coupling between the amplitude of $HFO_{\geq f \text{ Hz}}$ and the phase of slow-wave_{s Hz}. $HFO_{HIL_{\geq 80 \text{ Hz}}}$ denotes the rate of high-frequency oscillation defined by the Hilbert method. We could not compute z-score normalized $HFO_{HIL_{\geq 150 \text{ Hz}}}$ because the regression model failed to fit the data of $HFO_{HIL_{\geq 150 \text{ Hz}}}$ rates in a meaningful manner, due to the infrequent occurrence of events.

Additionally, we have generated new movie files (**Supplementary Movies 6, 7, and 8**), which allow readers to understand the expected range of normative MI and HFO_{HIL}>80 Hz rates at given cortical sites for different ages. Below, snapshots from **Supplementary Movies 6, 7 and 8** are provided.

Supplementary Movie 6. The normative ranges of cortical MI_{≥80} Hz and 0.5-1 Hz and MI_{≥80} Hz and 3-4 Hz.

On the left side, the video presents the mean of normative MI at given mesh points, as estimated by the univariate regression model incorporating $\sqrt{\text{age}}$. On the right side, it presents the mean plus two standard deviations, likewise estimated by the univariate regression model. For children who are 'n' years old, we computed the standard deviation across children between 'n' and 'n+3.9' years old. The brain images in this movie were created using FreeSurfer (<https://surfer.nmr.mgh.harvard.edu/fswiki/CorticalParcellation>).

Timeline

00:00-00:25 MI_{>80} Hz and 0.5-1 Hz.

00:25-00:50 MI_{>80} Hz and 3-4 Hz.

Supplementary Movie 7. The normative ranges of cortical $MI \geq 150$ Hz and 0.5-1 Hz and $MI \geq 150$ Hz and 3-4 Hz. On the left side, the video presents the mean of normative MI at given mesh points, as estimated by the univariate regression model incorporating $\sqrt{\text{age}}$. On the right side, it presents the mean plus two standard deviations, likewise estimated by the univariate regression model. For children who are 'n' years old, we computed the standard deviation across children between 'n' and 'n+3.9' years old. The brain images in this movie were created using FreeSurfer (<https://surfer.nmr.mgh.harvard.edu/fswiki/CorticalParcellation>).

Timeline

00:00-00:25 $MI > 150$ Hz and 0.5-1 Hz.

00:25-00:50 $MI > 150$ Hz and 3-4 Hz.

Supplementary Movie 8. The normative range of cortical HFOHIL ≥ 80 Hz. On the left side, the video presents the mean of normative HFOHIL >80 Hz at given mesh points, as estimated by the univariate regression model incorporating $\sqrt{\text{age}}$. On the right side, it presents the mean plus two standard deviations, likewise estimated by the univariate regression model. For children who are 'n' years old, we computed the standard deviation across children between 'n' and 'n+3.9' years old. The brain images in this movie were created using FreeSurfer (<https://surfer.nmr.mgh.harvard.edu/fswiki/CorticalParcellation>).

In the Discussion section, we have provided the following statements: “In the present study, we have provided atlases presenting the expected mean plus two standard deviations of MI biomarkers to illustrate the typical ranges for given age groups (**Supplementary Movies 6 and 7**). We believe our atlas has the potential to serve as a valuable reference in presurgical evaluations, as z-score normalized MI values were significantly higher in the SOZ compared to nonepileptic sites (**Table 4**). Compared to MI >80 Hz & 0.5-1 Hz, MI >80 Hz & 3-4 Hz showed much larger effect sizes of difference between the SOZ and nonepileptic sites, as suggested by the mixed model estimate (**Table 4**). This observation can be attributed to the notion that interictal epileptiform discharges are generally associated with a transient increase in HFO stereotypically coupled with a delta wave at 3-4 Hz⁶, whereas physiological HFO cycles between augmentation and attenuation during slow-wave sleep at <1 Hz¹⁸⁻²⁰. A prospective study is warranted to investigate whether a more inclusive resection of cortical sites, whose MI >80 Hz & 3-4 Hz values deviate from the age-specific normal range, would predict better postoperative seizure control in young children.”

“**Supplementary Movie 8** presents the mean plus two standard deviations of the HFO_{HIL} >80 Hz rate, as suggested by the regression model. This atlas may be useful in helping investigators understand the typical range of this biomarker value at given sites and given ages. Our mixed model analysis demonstrated that, within each brain lobe, z-score normalized HFO_{HIL} >80 Hz was higher in the SOZ compared to nonepileptic sites (**Table 4**).”

In the Methods section, we have provided the following paragraph: “**MI and HFO in the seizure onset zone (SOZ)** We employed mixed model analyses in given brain lobes to determine whether the SOZ electrode sites exhibited significantly higher MI and HFO rates compared to nonepileptic sites. Here, the dependent variable was z-score normalized $MI_{\geq 80 \text{ Hz and } 0.5-1 \text{ Hz}}$, $MI_{\geq 80 \text{ Hz and } 3-4 \text{ Hz}}$, $MI_{\geq 150 \text{ Hz and } 0.5-1 \text{ Hz}}$, $MI_{\geq 150 \text{ Hz and } 3-4 \text{ Hz}}$, $HFO_{HIL \geq 80 \text{ Hz}}$, and $HFO_{HIL \geq 150 \text{ Hz}}$ at a given electrode site. We computed a z-score normalized value of a given iEEG biomarker using the mean and standard deviation across 30 nonepileptic electrode sites closest to a given electrode contact (Kuroda et al., 2021); thereby, for children who are 'n' years old, we computed the mean and standard deviation across children between 'n' and 'n+3.9' years old. The fixed effect predictor included the SOZ label (SOZ = 1; nonepileptic = 0). The random effect factors included the intercept and patient. We deemed an FDR-corrected p-value of 0.05 (for six comparisons: six iEEG measures) as the significance threshold.

We have illustrated the mean plus two standard deviations of the aforementioned iEEG biomarkers, as indicated by the regression model, to help readers understand the typical ranges of these markers for given age groups (**Supplementary Movies 6-8**).”

[Comment 21] the lack of evidence supporting the hypothesis that visual memory consolidation and the strengthen of the MI in the occipital lobe are linked. I suggest dedicating more place in the paper to the neurocognitive aspect and the hypothesis possibly associated with this finding than to the need of improving the epileptogenic biomarkers. 1. It will be of interest to compare the MI in the “physiological” occipital lobe with the MI in the “pathological” ones and possibly with neuropsychological evaluations. Does it exist a “threshold” of MI that distinguishes the physiological from pathological channels? Do neurophysiological data correlate with neuropsychological examination? 2. If MI is involved in visual memory consolidation, children with visual memory impairment may possibly show modifications in this index. I think more data is needed to support your hypothesis on visual memory.

[Response 21] We sincerely thank Reviewer #3 for the valuable and constructive feedback. We concur with Reviewer #3's perspective, highlighting the necessity to conduct an experiment incorporating a pre-sleep visual task and a subsequent behavioral test upon awakening.

Several previous studies, including Landsness et al. (Sleep, 2009) with 12 healthy participants using scalp EEG, Yotsumoto et al. (Curr Biol, 2009) involving 21 healthy participants using fMRI, and Zhang et al. (Nat Commun, 2018) studying 12 patients through invasive EEG sampling predominantly from the temporal lobe regions, have employed suitable experimental designs to determine the significance of EEG biomarkers in visual memory consolidation. Landsness et al. (2009) illustrated that the performance of a visuo-motor task post-awakening was adversely affected by the acoustic stimuli-induced suppression of sleep slow waves. Yotsumoto et al. (2009) observed a correlation between the enhanced performance in a task involving visual perceptual learning and the level of trained-region-specific hemodynamic activation in the primary visual cortex. Zhang et al. (2018) found that the gamma activity_{30-90 Hz} in the temporal lobe, when replayed during sleep, was linked with successful recall upon awakening. By employing such a suitable study design, one could potentially confirm a causal link between visual memory consolidation and occipital MI, capable of rating delta-nested HFO during sleep.

In our study, however, none of our study patients were given a pre-sleep visual task followed by a behavioral test upon awakening. Thus, we were only able to assess the correlation between the z-score normalized occipital MI recorded intracranially and the baseline

neuropsychological data compiled before iEEG recording. We employed an additional mixed model analysis applied to 48 patients who underwent Peabody Picture Vocabulary Test, and this analysis failed to find a significant association between z-score normalized occipital MI and neuropsychological score.

In response to [Comment 33], our revised manuscript has moderated the intensity of the assertion that occipital delta-nested HFO assists visual memory consolidation throughout the main text. For example, we have removed the statements regarding visual memory consolidation from the abstract. We have also underscored in the discussion that the causal relationship between occipital delta-nested HFO and visual memory consolidation remains a hypothesis for future testing.

In the Results section, we have provided the following paragraph: “***Relationship between MI and neuropsychological score*** A total of 50 patients underwent Peabody Picture Vocabulary Test (PPVT) (Dunn and Dunn, 2007) prior to surgery; 48 out of the 50 patients had iEEG sampling from the occipital lobe, so they were included in the following statistical analysis. Mixed model analysis failed to show a significant association between z-score normalized MI_{>80 Hz & 0.5-1 Hz} and PPVT score (mixed model effect: 2.4×10^{-9} ; uncorrected p-value: 1.00; t-value: 4.4×10^{-5} ; DF: 453).”

In the Discussion section, we have provided the following paragraph: “In the current study, we found no significant association between the z-score normalized MI_{>80 Hz & 0.5-1 Hz} in the occipital lobe and the age-normalized PPVT score, an instrument measuring visual object recognition. Given that we failed to find an association between MI and PPVT, one must be cautious in extrapolating a causal relationship between delta-nested HFO and visual memory consolidation. Previous research has employed more targeted methodologies to discern the impact of EEG measures on visual memory consolidation (Landsness et al., 2009; Yotsumoto et al., 2009; Zhang et al., 2018). These studies incorporated a pre-sleep visual task, followed by a behavioral test upon awakening—a process that may more effectively reveal the causal relationship between cortical signals and visual memory consolidation. A study of 12 patients through invasive EEG sampling predominantly from the temporal lobe regions reported that gamma activity_{30-90 Hz} in the temporal lobe, when replayed during sleep, was associated with successful recall upon awakening (Zhang et al., 2018). Conversely, our study did not provide any of the participating patients with a pre-sleep visual task or a post-sleep behavioral test. A study with a proper visual perceptual task (Yotsumoto et al., 2009) and behavioral assessment before and after sleep is necessary to better determine the causal relationship between occipital delta-nested HFO and visual memory consolidation.”

In the Methods section, we have provided the following paragraph: “***Relationship between MI and neuropsychological score*** To explore the relationship between occipital MI and visual object recognition skills, we employed mixed model analysis on the patients who had undergone the PPVT (Dunn and Dunn, 2007) before surgery and had at least one nonepileptic electrode site within the occipital lobe. The dependent variable was the baseline PPVT score, and the fixed effect predictors were: [1] age at surgery (e.g., $\sqrt{\text{year}}$), [2] sex (female = 1), [3] presence of SOZ in the occipital lobe (yes = 1), [4] presence of MRI-visible structural lesion (yes = 1), [5] sampled hemisphere (left = 1), [6] number of oral antiseizure medications taken immediately before the initiation of iEEG recording, and [7] z-score normalized MI_{>80 Hz & 0.5-1 Hz}. The random effect factors included intercept and patient.”

[Comment 22] I found it limiting not to define better the criteria of "slow wave sleep". In animals, there is no clear consensus but in humans, it should correspond to the criteria of slow wave sleep N3 according to the AASM (American academy of sleep medicine) and specify if it is the first or second part of the night at least because the proportion of slow waves is not the same. It would also be necessary to specify on which way they determined the stage of sleep, knowing that the amplitude of the slow waves can be different according to the regions.

[Response 22] In the Method section, we detail our criteria for selecting the 20-minute slow-wave sleep epochs to be analyzed in this study: "We visually identified and analyzed the earliest available, artifact-free, 20-minute slow-wave sleep iEEG epochs that were at least 2 hours apart from ictal events (Bagshaw et al., 2009; Dümpelmann et al., 2015). We ensured that slow-wave sleep iEEG signals at non-epileptic sites displayed low-frequency (< 2 Hz) activity for at least 20% of the time during each 30-second epoch. We used Rayleigh's test to examine if the distribution of the onset time of the analyzed slow-wave sleep epochs deviated from a uniform distribution. Furthermore, we employed Spearman's rank test to investigate any potential correlation between patient age and the onset time of the analyzed slow-wave sleep epoch."

In our Results section, we provided the following statement: "The Rayleigh's test indicated a significant deviation from a uniform distribution in the onset time of studied slow-wave sleep epochs ($z=83.5$; $p<0.001$), with the peak onset time observed at 1:00 am. We found no significant correlation between patient age and the onset time of the studied slow-wave sleep epoch ($p=0.40$ using Spearman's rank test)."

Our methods are aligned with those of Bagshaw et al., *Epilepsia* 2009 and Dümpelmann et al., *Epilepsia* 2015, who visually assessed low-frequency activity on iEEG electrodes and defined slow-wave sleep based on the AASM criteria (Iber et al., 2007). Dümpelmann et al. (2015) indicated their analysis included iEEG signals during sleep stages 3 and 4. Our analytic approach aligns with the objective of selecting stable periods of sleep with minimal arousals, which was explicitly reported in a previous iEEG study (Lambert et al., *Clin Neurophysiol* 2022). None of these iEEG studies specified whether slow-wave sleep epochs were chosen from the first or second half of the night. As patients underwent periodic assessments of consciousness and vital signs during extraoperative iEEG recording, it wasn't feasible to select the analysis epoch from a specific part of the night. Nonetheless, our statistical findings, as mentioned above, demonstrate no significant systematic bias in our selected iEEG analysis period.

[Comment 23] Data are collected during sleep; do you think that dreams rather than the sleep modifications during this early stage of life can play a role? Did you look to spindles activities? Could be MI involved in such a change? Occipito-parietal regions are also taught to be involved in consciousness (Siclari et al., *Nat Neurosci* 2017), do you think your findings could be somehow linked to such experiences in the young child?

[Response 23] We greatly appreciate Reviewer #3's thoughtful and stimulating comments. In this study, we did not take into account the potential impact of dreaming during non-REM sleep (Siclari et al., *Nat Neurosci* 2017). We did not query our patients about their dreams immediately after arousal, hence we cannot speculate on the role of dreaming in our study. Acquiring reliable data from young children presents a significant challenge. As suggested by Reviewer #3, dreams may indeed play a crucial role in the development of sensorimotor and cognitive function. We must devise a study design that enables us to investigate the mechanism of dreaming utilizing intracranial EEG.

In our previous iEEG study, we found that saccadic eye movements were associated with a transient reduction and subsequent augmentation of HFO in the primary visual cortex during wakefulness and REM sleep (Uematsu et al., Neuroimage 2013). However, as we did not collect dream data from any of the patients, we were unable to determine the electrocorticographic correlates of dreaming.

We have been aware that sleep spindles can be recorded with iEEG (Asano et al, Clin Neurophysiol 2004). While we visually identified sleep spindles on iEEG in the present study, we did not perform a quantitative analysis on them. Our quantitative analysis focused on the coupling between HFO and slow waves ranging from 0.5 to 8 Hz, thus sleep spindles were not within the purview of this study.

Consequently, we have included the following statement in the Discussion: “We did not ask any of the patients about dream contents immediately following arousal; therefore, we are unable to determine whether developmental changes in MI and HFO are attributed to the change in dreaming behaviors during slow-wave sleep (Siclari et al., Nat Neurosci 2017).”

[Comment 24] From the videos, it seems that the mesial part of the fronto-parietal lobes behaves differently from the lateral ones, did you try to separate these areas in your statistical analysis? In particular, hippocampus is known to have physiological ripples especially during sleep involved in memory processing, so maybe it deserves a different approach that being included in all the temporal lobe.

[Response 24] To address Reviewer #3’s concern about **Supplementary Movie 6** showing the developmental changes of $MI > 80 \text{ Hz} \ \& \ 0.5\text{-}1 \text{ Hz}$, we performed ancillary mixed model analysis at each of the 22 regions of interest (ROIs). We provide the results of mixed model analyses in **Supplementary Tables 12-33**. We have verified that the results of these analyses do not change our conclusions.

We specifically found that the medial occipital region (mixed model estimate: 0.057/Year; uncorrected p-value: 0.0018; **Supplementary Table 17**), lateral occipital region (mixed model estimate: 0.042/Year; uncorrected p-value: 0.0027; **Supplementary Table 16**), and middle temporal region (mixed model estimate: 0.013/Year; uncorrected p-value: 0.0038; **Supplementary Table 19**) showed significant 'Age-dependent increase in $MI > 80 \text{ Hz} \ \& \ 0.5\text{-}1 \text{ Hz}$. No other ROIs (including the medial temporal region [mixed model estimate: 0.012/Year; uncorrected p-value: 0.086; **Supplementary Table 18**] or paracentral lobule in the medial frontal-parietal region [mixed model estimate: 0.017/Year; uncorrected p-value: 0.22; **Supplementary Table 21**]) showed significant 'Age-dependent modulation in $MI > 80 \text{ Hz} \ \& \ 0.5\text{-}1 \text{ Hz}$.

[Comment 25] The authors share with the community all data and analysis code permitting reproducibility of the results. Methods are well described and signal analysis is rigorous, with a comparison of different methods of HFOs detections (open source: ripple lab). The match of neurophysiological findings with MRI DWI improve the quality of the results and represent an important step towards a multidisciplinary approach to physiology. However, a more detailed methodological analysis should be done and it is not in my expertise (in particular MRI tractography analysis and statistical analysis).

[Response 25] We have provided the exact number of tracts connecting pairs of mesh points showing a significant relationship between 'Age and iEEG biomarkers as follows.

“White matter tracts connecting cortices with developmental MI co-growth Univariate regression models, incorporating $\sqrt{\text{age}}$ as an independent variable, found that 159 out of the 9464 cortical mesh points (1.7%) showed significantly positive regression slopes, whereas no cortical mesh points showed significantly negative regression slopes. DWI analysis revealed that 2 tracts including the vertical occipital fasciculi and posterior callosal fibers directly connected mesh point pairs showing significant developmental co-growth of $\text{MI}_{\geq 80 \text{ Hz}} \& 0.5\text{-}1 \text{ Hz}$ (**Figure 5** and **Supplementary Movie 4**).”

“White matter tracts connecting cortices with developmental HFO co-diminution Univariate regression models incorporating $\sqrt{\text{age}}$ as an independent variable found that 902 out of the 9464 cortical mesh points (9.5%) showed significantly negative regression slopes, whereas no cortical mesh points showed significantly positive regression slopes. DWI analysis revealed that 636 tracts, including the arcuate fasciculus, corpus callosum, extreme capsule, frontal aslant tract, inferior fronto-occipital fasciculus, inferior longitudinal fasciculus, middle longitudinal fasciculus, and superior longitudinal fasciculus, directly connected mesh point pairs showing significant developmental co-diminution of $\text{HFO}_{\text{HIL} \geq 80 \text{ Hz}}$ rate (**Figure 5** and **Supplementary Movie 5**).”

[Comment 26] Limitations are represented by the subdural iEEG technique that evaluates more the cortical aspects of the brain, without a good sampling i.e. of the mesial temporal regions that are fundamental for memory consolidation; this limits to me the word “global atlas” because some important regions are not sufficiently sampled.

[Response 26] We have removed the term: “whole-brain level” from the manuscript.

[Comment 27] Minor comments I could not find the ancillary analysis of the rates of $\text{HFO} > 150 \text{ Hz}$ and 250 Hz that could indeed be of interest regarding the difference between pathological and physiological HFO even if done on a smaller cohort of patients.

[Response 27] We have decided not to include the analysis of $\text{HFO}_{\geq 250 \text{ Hz}}$, as requested by Reviewers #1 and #2, due to the sampling rate employed in the present study. Nevertheless, we have presented the results of the mixed model analysis comparing $\text{MI}_{\geq 150 \text{ Hz}}$ & slow waves between SOZ and non-epileptic sites, as detailed in **Table 4**.

Due to the infrequent occurrence of events, we were unable to compute the statistical deviation of $\text{HFO}_{\text{HIL} \geq 150 \text{ Hz}}$ in a meaningful manner.

[Comment 28] Regarding occipital HFA independently of epileptogenic area, you can cite also Melani 2013 Continuous High Frequency Activity: A peculiar SEEG pattern related to specific brain regions.

[Response 28] We have cited Melani et al., Clin Neurophysiol 2013, as shown below: “ $\text{HFO}_{\text{MNI} \geq 80 \text{ Hz}}$ rate was very low in the occipital lobes throughout all ages; this MNI method-specific observation can be attributed to its detector being designed to be agnostic to persistent forms of high-frequency activity (Zelmann et al., 2010), as often seen in the nonepileptic occipital lobe (Nagasawa et al., 2012; Melani et al., 2013).”

[Comment 29] What do you mean with the word “enhanced MI?” could you better clarify? [Response 29] We have replaced “enhanced MI” with “higher MI”.

Reviewer #4 (Remarks to the Author):

[Comment 30] In this paper, the authors develop a normative atlas for phase-amplitude coupling (via the modulation index, MI) and high frequency oscillations (HFO) in 114 pediatric patients implanted with iEEG. They report several significant relationships between age and MI, as well as age and HFOs, and they combine this with white matter tractography, which provides evidence of connectivity between the associated regions. The large dataset, the testing of multiple frequency band pairs for MI, and the use of four different HFO detectors are strengths. The focus on a pediatric patient population and the combination of computational iEEG analysis with white matter tractography are novel. However, I have several significant methodological concerns, as detailed below.

[Response 30] We appreciate the constructive comments provided by Reviewer #4. We have made a diligent effort to address each comment raised by Reviewer #4.

[Comment 31] The inclusion of only four (out of 114) patients above 20 years old is concerning, as the three 30-40 year-old-patients are outliers in terms of age and may have an outsized influence on the regression results. Because the study focuses on a pediatric patient population, it seems the authors would be justified in excluding these adult subjects. At the very least, I would suggest that the authors redo all regression results after excluding those four subjects, to verify that the significant results still hold.

[Response 31] We performed ancillary mixed model analyses, excluding three patients of 21 years old and above. We have verified that the results of these analyses do not change our conclusions.

For example, as shown in **Table 2 and Supplementary Table 2** below, the developmental growth (slope/ $\sqrt{\text{year}}$) of MI was prominent in the occipital lobe and modest if any in the remaining brain lobes.

Lobe	Slope (/ $\sqrt{\text{year}}$)	Uncorrected p-value	t-value	DF	Lower 95% CI	Upper 95% CI
Frontal	7.0×10^{-3}	$*3.0 \times 10^{-6}$	4.7	2978	4.1×10^{-3}	9.9×10^{-3}
Temporal	7.2×10^{-3}	$*6.5 \times 10^{-8}$	5.4	2393	4.6×10^{-3}	9.7×10^{-3}
Parietal	0.011	$*2.4 \times 10^{-9}$	6.0	1999	7.4×10^{-3}	0.015
Occipital	0.047	$*3.1 \times 10^{-12}$	7.1	873	0.034	0.060

Table 2. The results of regression analysis to assess the effect of $\sqrt{\text{age}}$ on $\text{MI}_{\geq 80 \text{ Hz}} \& 0.5\text{-}1 \text{ Hz}$. $\text{MI}_{\geq 80 \text{ Hz}}$ and $0.5\text{-}1 \text{ Hz}$ denotes the strength of coupling between the amplitude of high-frequency oscillation $>80 \text{ Hz}$ and the phase of slow-wave $0.5\text{-}1 \text{ Hz}$. CI: confidence interval. DF: degree of freedom. *: significant with False Discovery Rate (FDR) correction. **Supplementary Table 2** presents the results of the analysis, excluding three patients of 21 years old and above.

Lobe	Slope (/ $\sqrt{\text{year}}$)	Uncorrected p-value	t-value	DF	Lower 95% CI	Upper 95% CI
Frontal	5.8×10^{-3}	* 6.9×10^{-4}	3.4	2895	2.5×10^{-3}	9.2×10^{-3}
Temporal	5.1×10^{-3}	* 3.0×10^{-4}	3.6	2314	2.4×10^{-3}	7.9×10^{-3}
Parietal	8.2×10^{-3}	* 3.0×10^{-5}	4.2	1949	4.4×10^{-3}	0.012
Occipital	0.049	* 4.1×10^{-11}	6.7	831	0.034	0.063

Supplementary Table 2. The results of ancillary regression analysis to assess the effect of 'Iage on MI $_{>80 \text{ Hz}}$ & 0.5-1 Hz in a given brain lobe. Here, we present the results of ancillary analyses, excluding three patients of 21 years old and above. MI $_{>80 \text{ Hz}}$ and 0.5-1 Hz denotes the strength of coupling between the amplitude of high-frequency oscillation $_{>80 \text{ Hz}}$ and the phase of slow-wave $_{0.5-1 \text{ Hz}}$. CI: confidence interval. DF: degree of freedom. *: significant with False Discovery Rate (FDR) correction.

We also provided the results of mixed model analyses, with all 114 patients included (**Supplementary Tables 4-7**) and with three patients of >21 years old excluded (**Supplementary Tables 8-11**). Both models commonly indicated that the effect of $\sqrt{\text{age}}$ on MI was significant in the occipital lobe but not in the remaining brain lobes.

As shown in **Table 3 and Supplementary Table 3** below, the developmental diminution (slope/ $\sqrt{\text{year}}$) of HFO $_{\text{HIL } >80 \text{ Hz}}$ rate was prominent in the frontal, temporal, and parietal lobes, regardless of excluding three patients of 21 years old and above.

Lobe	Slope (/ $\sqrt{\text{year}}$)	Uncorrected p-value	t-value	DF	Lower 95% CI	Upper 95% CI
Frontal	-0.41	* 4.3×10^{-54}	-15.8	2978	-0.46	-0.36
Temporal	-0.30	* 7.4×10^{-28}	-11.0	2393	-0.36	-0.25
Parietal	-0.33	* 3.7×10^{-21}	-9.5	1999	-0.40	-0.26
Occipital	0.081	0.31	1.0	873	-0.075	0.24

Table 3. The results of regression analysis to assess the effect of 'Iage on HFO $_{\text{HIL } \geq 80 \text{ Hz}}$ occurrence rate. HFO $_{\text{HIL } >80 \text{ Hz}}$: high-frequency oscillation $_{>80 \text{ Hz}}$ defined by the Hilbert method. CI: confidence interval. DF: degree of freedom. *: significant with False Discovery Rate (FDR) correction. **Supplementary Table 3** presents the results of the analysis, excluding three patients of 21 years old and above.

Lobe	Slope (/ $\sqrt{\text{year}}$)	Uncorrected p-value	t-value	DF	Lower 95% CI	Upper 95% CI
Frontal	-0.45	* 2.7×10^{-49}	-15.0	2895	-0.51	-0.39
Temporal	-0.38	* 1.6×10^{-34}	-12.5	2314	-0.44	-0.32
Parietal	-0.39	* 4.9×10^{-25}	-10.5	1949	-0.46	-0.32
Occipital	0.13	0.15	1.4	831	-0.046	0.30

Supplementary Table 3. The results of ancillary regression analysis to assess the effect of $\sqrt{\text{age}}$ on HFO_{HIL} ≥ 80 Hz occurrence rate in a given brain lobe. Here, we present the results of ancillary analyses, excluding three patients of 21 years old and above. HFO_{HIL} > 80 Hz: high-frequency oscillation > 80 Hz defined by the Hilbert method. CI: confidence interval. DF: degree of freedom. *: significant with False Discovery Rate (FDR) correction.

We also provided the results of mixed model analyses, with all 114 patients included (**Supplementary Tables 34-37**) and with three patients of > 21 years old excluded (**Supplementary Tables 38-41**). Both models commonly indicated that the developmental diminution of HFO_{HIL} > 80 Hz rate was significant in the frontal, temporal, and parietal lobes but not in the occipital lobe.

[Comment 32] 2) It is difficult to see evidence of the reported changes in MI and HFOs with age, as shown in Figure 2. I appreciate that all data points are shown, but there are so many data points and the figures are so small that the relationships with age are not evident. Perhaps converting these to violin plots or boxplots (individual plots for each year of age, or each two years of age) would be more convincing, so the reader could better interpret the distribution of values at each age?

[Response 32] In **Figure 2** and **Supplementary Figure 3**, we have illustrated the data distribution using violin plots as shown below.

Figure 2. The developmental changes of cortical MI and HFO at given lobes. a $MI_{\geq 80 \text{ Hz} \& 0.5-1 \text{ Hz}}$ denotes the strength of coupling between the amplitude of $HFO_{\geq 80 \text{ Hz}}$ and the phase of slow-wave $0.5-1 \text{ Hz}$, as rated by modulation index. **b** Occurrence rate (/min) of $HFO_{HIL \geq 80 \text{ Hz}}$ as defined by the Hilbert method. In each violin plot, a regression line is provided based on a model incorporating the square root of age ($\sqrt{\text{age}}$) as an independent variable. The white circle within each violin plot represents the median.

Supplementary Figure 3. The developmental changes of cortical MI and HFO at given lobes. **a** $MI_{280\text{ Hz} \& \text{3-4 Hz}}$: the strength of coupling between the amplitude of $HFO_{280\text{ Hz}}$ and the phase of slow-wave_{3-4 Hz}, as rated by modulation index. **b** $HFO_{STE\ 280\text{ Hz}}$ occurrence rate (/min). **c** $HFO_{SLL\ 280\text{ Hz}}$ occurrence rate. **d** $HFO_{MNI\ 280\text{ Hz}}$ occurrence rate. In each violin plot, a regression line is provided based on a model incorporating the square root of age ($\sqrt{\text{age}}$) as an independent variable. The white circle within each violin plot represents the median. MI: modulation index. HFO_{STE} : high-frequency oscillation (HFO) defined by Staba et al., 2002. HFO_{SLL} : HFO defined by Gardner et al., 2007. HFO_{HIL} : HFO defined by Crépon et al., 2010. HFO_{MNI} : HFO defined by Zelmann et al., 2010.

Supplementary references.

- Staba, R. J., Wilson, C. L., Bragin, A., Fried, I. & Engel, J. Jr. Quantitative analysis of high-frequency oscillations (80-500 Hz) recorded in human epileptic hippocampus and entorhinal cortex. *J Neurophysiol.* **88**, 1743-1752 (2002).
- Gardner, A. B., Worrell, G. A., Marsh, E., Dlugos, D. & Litt, B. Human and automated detection of high-frequency oscillations in clinical intracranial EEG recordings. *Clin Neurophysiol.* **118**, 1134-1143 (2007).
- Crépon, B. et al. Mapping interictal oscillations greater than 200 Hz recorded with intracranial macroelectrodes in human epilepsy. *Brain.* **133**, 33-45 (2010).
- Zelmann, R. et al. Automatic detector of high frequency oscillations for human recordings with macroelectrodes. *Annu Int Conf IEEE Eng Med Biol Soc.* **2010**, 2329-2333 (2010).

[Comment 33] Minor comments: The WinPACT toolbox can calculate phase-amplitude coupling using several different methods. Which one was used here?

[Response 33] The screenshot below presents the method and parameters utilized in our analysis of modulation index. To export Canolty's modulation index values, we employed the

EEG.etc.winPACT.canoltysMIAllChan command within the winPACT toolbox (<https://github.com/scn/winPACT>).

In the revised manuscript, we provided the following statement: “We employed the EEG.etc.winPACT.canoltysMIAllChan command within the winPACT toolbox (<https://github.com/scn/winPACT>).”

[Comment 34] The first two paragraphs on p.18 seem to repeat the same information and could be combined into one paragraph.

[Response 34] We have minimized the repetition of the same information within the main text, as suggested by Reviewer #4.

[Comment 35] The data in Figure 5A are difficult to see, as the figures are very dark.

[Response 35] We have improved the visibility of white matter tracts presented in **Figure 5**, as shown below.

Figure 5. Dynamic tractography. The video snapshots present the varying intensity of **a** co-growth of $MI \geq 80$ Hz & 0.5-1 Hz and **b** co-diminution of $HFO_{HIL} \geq 80$ Hz at ages 1, 5, 10, and 20 years, as estimated by univariate regression analysis incorporating 'Iage as an independent variable. **Supplementary Movies 4-5** show the data across generations from 1 to 21 years. The brain images in this figure were created using FreeSurfer (<https://surfer.nmr.mgh.harvard.edu/fswiki/CorticalParcellation>).

[Comment 36] Some clarification on the white matter tract results (p.21-22) would be helpful. My interpretation is that a total of 9464 mesh points were tested across the entire brain, and 159 showed a significant relationship between the square root of age and the MI value. Some of these 159 points were deemed to be connected based on the DWI analysis; it would be helpful if the authors gave the exact numbers of pairs/connections. I have the same question for the 902 mesh points associated with HFO co-diminution.

[Response 36] We have provided the exact number of tracts connecting pairs of mesh points showing a significant relationship between 'Iage and iEEG biomarkers as follows.

“White matter tracts connecting cortices with developmental MI co-growth Univariate regression models, incorporating 'Iage as an independent variable, found that 159 out of the 9464 cortical mesh points (1.7%) showed significantly positive regression slopes, whereas no cortical mesh points showed significantly negative regression slopes. DWI analysis revealed that 2 tracts including the vertical occipital fasciculi and posterior callosal fibers directly connected mesh point pairs showing significant developmental co-growth of $MI \geq 80$ Hz & 0.5-1 Hz (**Figure 5** and **Supplementary Movie 4**).”

“White matter tracts connecting cortices with developmental HFO co-diminution Univariate regression models incorporating 'Iage as an independent variable found that 902 out of the 9464 cortical mesh points (9.5%) showed significantly negative regression slopes, whereas no cortical mesh points showed significantly positive regression slopes. DWI analysis revealed that 636 tracts,

including the arcuate fasciculus, corpus callosum, extreme capsule, frontal aslant tract, inferior fronto-occipital fasciculus, inferior longitudinal fasciculus, middle longitudinal fasciculus, and superior longitudinal fasciculus, directly connected mesh point pairs showing significant developmental co-diminution of HFO_{HIL}≥80 Hz rate (**Figure 5** and **Supplementary Movie 5**).”

[Comment 37] In the first paragraph of the discussion, it would be helpful to know what a typical MI value is in epileptogenic cortex, especially in the occipital lobe, if it is known. I am wondering if the high occipital lobe values in non-epileptogenic cortex are really large enough to be mistaken as pathological.

Similarly, in the second section of the discussion (on clinical significance of the HFO results), it would be helpful to know how the ranges of HFO rates reported here compare to the adult normative atlases that have been developed. Do the values for the older children approach those of adults?

[Response 37] We have included new movie files, which present the mean as well as the mean + 2 standard deviations of MI_{≥80} Hz & slow waves (**Supplementary Movie 6**), MI_{≥150} Hz & slow waves (**Supplementary Movie 7**), and HFO_{HIL}≥80 Hz rate (**Supplementary Movie 8**). These videos provide valuable insights to investigators, illustrating the anticipated magnitude of values in nonepileptic cortices in a given patient age. Below, we have presented the snapshots of movie files.

Supplementary Movie 6. The normative ranges of cortical MI_{≥80} Hz and 0.5-1 Hz and MI_{≥80} Hz and 3-4 Hz. On the left side, the video presents the mean of normative MI at given mesh points, as estimated by the univariate regression model incorporating $\sqrt{\text{age}}$. On the right side, it presents the mean plus two standard deviations, likewise estimated by the univariate regression model. For children who are 'n' years old, we computed the standard deviation across children between 'n' and 'n+3.9' years old. The brain images in this movie were created using FreeSurfer (<https://surfer.nmr.mgh.harvard.edu/fswiki/CorticalParcellation>).

Timeline

00:00-00:25 MI>80 Hz and 0.5-1 Hz.

00:25-00:50 MI>80 Hz and 3-4 Hz.

Supplementary Movie 7. The normative ranges of cortical MI \geq 150 Hz and 0.5-1 Hz and MI \geq 150 Hz and 3-4 Hz.

On the left side, the video presents the mean of normative MI at given mesh points, as estimated by the univariate regression model incorporating $\sqrt{\text{age}}$. On the right side, it presents the mean plus two standard deviations, likewise estimated by the univariate regression model. For children who are 'n' years old, we computed the standard deviation across children between 'n' and 'n+3.9' years old. The brain images in this movie were created using FreeSurfer (<https://surfer.nmr.mgh.harvard.edu/fswiki/CorticalParcellation>).

Timeline

00:00-00:25 MI>150 Hz and 0.5-1 Hz.

00:25-00:50 MI>150 Hz and 3-4 Hz.

Supplementary Movie 8. The normative range of cortical HFO_{HIL} ≥80 Hz. On the left side, the video presents the mean of normative HFO_{HIL} >80 Hz at given mesh points, as estimated by the univariate regression model incorporating $\sqrt{\text{age}}$. On the right side, it presents the mean plus two standard deviations, likewise estimated by the univariate regression model. For children who are 'n' years old, we computed the standard deviation across children between 'n' and 'n+3.9' years old. The brain images in this movie were created using FreeSurfer (<https://surfer.nmr.mgh.harvard.edu/fswiki/CorticalParcellation>).

Furthermore, additional mixed model analysis revealed that electrode contacts located within the seizure onset zone (SOZ) exhibited significantly higher z-score normalized modulation index (MI) and HFO_{HIL} >80 Hz (**Table 4**). These findings provide support for the hypothesis that statistical deviations from normative means could aid in localizing the epileptogenic zone during presurgical evaluation of epilepsy.

Biomarker	Lobe	Mixed model estimate	SE	t-value	DF	p-value	Lower 95% CI	Upper 95% CI
MI>80 Hz and 0.5-1 Hz	Frontal	0.75	0.070	10.8	3238	< 0.001	0.61	0.89
	Temporal	3.00	0.48	6.25	2811	< 0.001	2.06	3.94
	Parietal	1.27	0.11	11.8	2242	< 0.001	1.06	1.48
	Occipital	0.45	0.14	3.17	958	0.0016	0.17	0.73
MI>80 Hz and 3-4 Hz	Frontal	3.52	0.18	19.7	3238	< 0.001	3.17	3.87
	Temporal	4.10	0.33	12.4	2811	< 0.001	3.45	4.75
	Parietal	5.82	0.30	19.1	2242	< 0.001	5.22	6.42

	Occipital	1.83	0.21	8.73	958	< 0.001	1.42	2.24
$MI_{\geq 150 \text{ Hz and } 0.5-1 \text{ Hz}}$	Frontal	0.51	0.055	9.30	3238	< 0.001	0.40	0.62
	Temporal	2.56	0.81	3.14	2811	0.0017	0.96	4.15
	Parietal	0.62	0.065	9.50	2242	< 0.001	0.49	0.75
	Occipital	0.079	0.12	0.67	958	0.50	-0.15	0.31
$MI_{\geq 150 \text{ Hz and } 3-4 \text{ Hz}}$	Frontal	3.70	0.30	12.5	3238	< 0.001	3.12	4.28
	Temporal	3.77	0.73	5.16	2811	< 0.001	2.34	5.20
	Parietal	5.41	0.43	12.5	2242	< 0.001	4.56	6.26
	Occipital	0.94	0.18	5.11	958	< 0.001	0.58	1.31
$HFO_{HIL \geq 80 \text{ Hz}}$	Frontal	2.45	0.079	30.9	3238	< 0.001	2.30	2.61
	Temporal	1.85	0.080	23.3	2811	< 0.001	1.70	2.01
	Parietal	2.95	0.10	28.7	2242	< 0.001	2.75	3.16
	Occipital	1.59	0.14	11.4	958	< 0.001	1.32	1.86

Table 4. Comparison of intracranial EEG biomarker values between the seizure onset and nonepileptic sites. The results of mixed model analyses are provided with uncorrected p-value. All z-score normalized biomarker values mentioned in this table (except $MI_{\geq 150 \text{ Hz and } 0.5-1 \text{ Hz}}$) were significantly higher in the seizure onset than in the nonepileptic sites. CI: 95% confidence interval. DF: degree of freedom. SE: standard error. $MI_{\geq f \text{ Hz and } s \text{ Hz}}$ denotes the strength of coupling between the amplitude of $HFO_{\geq f \text{ Hz}}$ and the phase of slow-wave_{s Hz}. $HFO_{HIL \geq 80 \text{ Hz}}$ denotes the rate of high-frequency oscillation defined by the Hilbert method. We could not compute z-score normalized $HFO_{HIL \geq 150 \text{ Hz}}$ because the regression model failed to fit the data of $HFO_{HIL \geq 150 \text{ Hz}}$ rates in a meaningful manner, due to the infrequent occurrence of events.

In the Discussion section, we have provided the following statements: “In the present study, we have provided atlases presenting the expected mean plus two standard deviations of MI biomarkers to illustrate the typical ranges for given age groups (**Supplementary Movies 6 and 7**). We believe our atlas has the potential to serve as a valuable reference in presurgical evaluations, as z-score normalized MI values were significantly higher in the SOZ compared to nonepileptic sites (**Table 4**). Compared to $MI_{\geq 80 \text{ Hz} \& 0.5-1 \text{ Hz}}$, $MI_{\geq 80 \text{ Hz} \& 3-4 \text{ Hz}}$ showed much larger effect sizes of difference between the SOZ and nonepileptic sites, as suggested by the mixed model estimate (**Table 4**). This observation can be attributed to the notion that interictal epileptiform discharges are generally associated with a transient increase in HFO stereotypically coupled with a delta wave at 3-4 Hz⁶, whereas physiological HFO cycles between augmentation and attenuation during slow-wave sleep at <1 Hz¹⁸⁻²⁰. A prospective study is warranted to investigate whether a more inclusive resection of cortical sites, whose $MI_{\geq 80 \text{ Hz} \& 3-4 \text{ Hz}}$ values deviate from the age-specific normal range, would predict better postoperative seizure control in young children.”

“**Supplementary Movie 8** presents the mean plus two standard deviations of the $HFO_{HIL \geq 80 \text{ Hz}}$ rate, as suggested by the regression model. This atlas may be useful in helping investigators understand the typical range of this biomarker value at given sites and given ages. Our mixed model analysis demonstrated that, within each brain lobe, z-score normalized $HFO_{HIL \geq 80 \text{ Hz}}$ was higher in the SOZ compared to nonepileptic sites (**Table 4**).”

In the Methods section, we have provided the following paragraph: “***MI and HFO in the seizure onset zone (SOZ)*** We employed mixed model analyses in given brain lobes to determine whether the SOZ electrode sites exhibited significantly higher MI and HFO rates compared to nonepileptic sites. Here, the dependent variable was z-score normalized $MI_{\geq 80 \text{ Hz and } 0.5-1 \text{ Hz}}$, $MI_{\geq 80 \text{ Hz and } 3-4 \text{ Hz}}$, $MI_{\geq 150 \text{ Hz and } 0.5-1 \text{ Hz}}$, $MI_{\geq 150 \text{ Hz and } 3-4 \text{ Hz}}$, $HFO_{HIL \geq 80 \text{ Hz}}$, and $HFO_{HIL \geq 150 \text{ Hz}}$ at a given electrode site. We computed a z-score normalized value of a given iEEG biomarker using the mean and standard deviation across 30 nonepileptic electrode sites closest to a given electrode contact (Kuroda et al., 2021); thereby, for children who are 'n' years old, we computed the mean and standard deviation across children between 'n' and 'n+3.9' years old. The fixed effect predictor included the SOZ label (SOZ = 1; nonepileptic = 0). The random effect factors included the intercept and patient. We deemed an FDR-corrected p-value of 0.05 (for six comparisons: six iEEG measures) as the significance threshold.

We have illustrated the mean plus two standard deviations of the aforementioned iEEG biomarkers, as indicated by the regression model, to help readers understand the typical ranges of these markers for given age groups (**Supplementary Movies 6-8**).”

Please note that we have revised our manuscript entirely so that it complies with the *Nature Communications* formatting instructions. The specific changes include: [1] reformatting the list of authors; [2] shortening the abstract to less than 150 words; [3] removal of subheadings from the introduction; [4] starting the final paragraph of the introduction with the phrase “In this study, our normative atlases reveal ...”; [5] removal of numbering from the subheading; [6] removal of subheadings from the discussion; [7] revision of the data availability statement; [8] revision of the code availability statement; [9] reformatting of reference citations; [10] reduction of the number of references to no more than 70; [11] reformatting of the author contributions; [12] reformatting of **Figures 1-5**; [13] reformatting of **Tables 1-4**; [14] reformatting of the supplementary information (**Supplementary Tables 1 – 41**, **Supplementary Figures 1 – 4**, and Legends for **Supplementary Movies 1 - 8**); [15] conversion of the supplementary information to a PDF document; [16] reformatting of the main text; and [17] reduction of the number of words in the main text (not including figure legends or Methods) to 6000.

REVIEWERS' COMMENTS

Reviewer #1 (Remarks to the Author):

The authors have replied sufficiently to all my comments.

Very nice manuscript.

Reviewer #2 (Remarks to the Author):

The authors responded to all my comments and adapted the manuscript accordingly. Much additional information was provided.

I have one remaining remark about the last paragraph of the introduction, which reads like a conclusion instead of a hypothesis to me?

Reviewer #3 (Remarks to the Author):

The authors responded to all my comments and the manuscript is now improved, I'm OK with this revised version

Reviewer #4 (Remarks to the Author):

The authors have very thoroughly addressed my comments and improved the paper. In particular, I appreciated the revised figures and the additional analysis of the SOZ channels.

There is one drawback to the thoroughness of the additional analysis, however; the results section has become challenging to read. I found myself needing to flip back and forth between different sections of the paper to compare (e.g., to compare the the regression results for HFOs to the mixed model results

for HFOs, or to compare results in the text to results in a table). I think the results could be streamlined to improve readability in two ways:

1) Many sections in the results contain long lists of statistical results, sometimes repeating information from existing tables. If these were all moved to tables, this would make the text easier to read.

2) If the MI results were grouped together into one section, or into sequential sections, it would be easier to see how all of these related results form a cohesive picture. Same for the HFO results.

I recognize that this may be a matter of personal preference, but I wanted to share my perspective as a reader, in case it is helpful for the authors.

REVIEWER COMMENTS

Reviewer #1 (Remarks to the Author):

Comment 1: The authors have replied sufficiently to all my comments. Very nice manuscript.

Response 1: We are pleased that Reviewer #1 is satisfied with our revised manuscript. We extend our gratitude once again.

Reviewer #2 (Remarks to the Author):

Comment 2: The authors responded to all my comments and adapted the manuscript accordingly. Much additional information was provided. I have one remaining remark about the last paragraph of the introduction, which reads like a conclusion instead of a hypothesis to me?

Response 2: We appreciate Reviewer #2's feedback. In accordance with the Nature Communications Format Instruction, the final paragraph of our introduction has been specifically structured. The guidelines recommend that the last paragraph commence with phrases such as "In this work" or "Here, we show", followed by a concise summary of the major results and conclusions in the present tense. As such, we would like to retain our current format. We hope this provides clarity on our decision and trust that Reviewer #2 will understand our adherence to the journal's guidelines.

Reviewer #3 (Remarks to the Author):

Comment 3: The authors responded to all my comments and the manuscript is now improved, I'm OK with this revised version.

Response 3: We are gratified to know that Dr. Pizzo approves of our revisions. Our heartfelt thanks are extended once more.

Reviewer #4 (Remarks to the Author):

Comment 4: The authors have very thoroughly addressed my comments and improved the paper. In particular, I appreciated the revised figures and the additional analysis of the SOZ channels. There is one drawback to the thoroughness of the additional analysis, however; the results section has become challenging to read. I found myself needing to flip back and forth between different sections of the paper to compare (e.g., to compare the the regression results for HFOs to the mixed model results for HFOs, or to compare results in the text to results in a table). I think the results could be streamlined to improve readability in two ways:

1) Many sections in the results contain long lists of statistical results, sometimes repeating information from existing tables. If these were all moved to tables, this would make the text easier to read.

Response 4: We appreciate Reviewer #4's insightful suggestion. In line with the Nature Portfolio Reporting Summary Guidelines, we aimed to provide comprehensive statistical parameter descriptions. However, we acknowledge that in doing so, we may have affected the manuscript's readability. To address this, we've reduced redundant descriptions when they match details already available in our tables. Given the journal's constraint of a maximum of 10 figure/table items in the main text, it is not feasible to capture all statistical results solely in tables. We trust that Reviewer #4 will understand our decision to remain compliant with the journal's guidelines.

We have made a specific revision on page 31 of the main text. The 95% confidence interval details, which are also found in **Table 3**, have been removed from the main text. The revised statement now reads: “The regression slope was -0.41/Year in the frontal lobe, -0.30/Year in the temporal lobe, -0.33/Year in the parietal lobe, and +0.081/Year in the occipital lobe (see the detailed statistical results in **Table 3**).”

Comment 5: 2) If the MI results were grouped together into one section, or into sequential sections, it would be easier to see how all of these related results form a cohesive picture. Same for the HFO results. I recognize that this may be a matter of personal preference, but I wanted to share my perspective as a reader, in case it is helpful for the authors.

Response 5: We wish to maintain the current sequence of result presentation. At present, our results are presented in the following order:

- Developmental changes of iEEG biomarkers assessed by regression model (MI vs. HFO);
- Developmental changes of iEEG biomarkers assessed by mixed model analysis (MI vs. HFO);
- Developmental changes of spectral frequency band of slow waves coupled with HFO;
- Developmental changes in connectivity via white matter tracts (MI vs. HFO);
- iEEG biomarker utility to localize the seizure onset zone (MI vs. HFO);
- iEEG biomarker utility to predict neuropsychological data (MI).

Preserving this sequence is essential to effectively contrast the developmental changes between MI and HFO.